# Fungal Als proteins hijack host death effector domains to promote inflammasome signaling

Tingting Zhou[1], Norma V. Solis[2], Michaela Marshall [3], Qing Yao[4,6], Eric Pearlman [3], Scott G. Filler [2,5] & Haoping Liu [1] ✉

High-damaging *Candida albicans* strains tend to form hyphae and exacerbate intestinal inflammation in ulcerative colitis patients through IL-1β-dependent mechanisms. Fungal agglutinin-like sequence (Als) proteins worsen DSS-induced colitis in mouse models. FADD and caspase-8 are important regulators of gut homeostasis and inflammation. However, whether they link directly to fungal proteins is not fully understood. Here, we report that Als proteins induce IL-1β release in immune cells. We show that hyphal Als3 is internalized in macrophages and interacts with caspase-8 and the inflammasome adaptor apoptosis-associated speck-like protein containing a CARD (ASC). Caspase-8 is essential for Als3-mediated ASC oligomerization and IL-1β processing. In non-immune cells, Als3 is associated with cell death core components FADD and caspase-8. N-terminal Als3 (N-Als3) expressed in Jurkat cells partially inhibits apoptosis. Mechanistically, N-Als3 promotes oligomerization of FADD and caspase-8 through their death effector domains (DEDs). N-Als3 variants with a mutation in the peptide-binding cavity or amyloid-forming region are impaired in DED oligomerization. Together, these results demonstrate that DEDs are intracellular sensors of Als3. This study identifies additional potential targets to control hypha-induced inflammation.

*Candida albicans* (*C. albicans*), an opportunistic yeast, is a member of the human mycobiota in the oral, gastrointestinal, and vaginal mucosa. In patients with ulcerative colitis, high-damaging *C. albicans* strains form hyphae and exacerbate intestinal inflammation[1]. In immuno-compromised and critically ill patients, *C. albicans* transitions from a commensal organism in the gut to an opportunistic pathogen that causes disseminated candidiasis, which is a leading cause of nosocomial bloodstream infections. In developed countries, this leads to a mortality rate that can exceed 40%, even in patients given antifungal treatment[2].

Hyphal morphogenesis controls the balance between gut commensalism and invasive infection[3–5]. *C. albicans* invades host tissues through a process associated with yeast-to-hyphal transition[6,7]. Hyphae induce a higher level of IL-1β secretion than the yeast form[8,9]. The presence of gut *C. albicans* strains with a high ability to induce IL-1β production in phagocytes is positively correlated with increased disease severity in their host with ulcerative colitis[1]. *C. albicans* candidalysin, a hypha-secreted pore-forming toxin, damages epithelial cells and fuels proinflammatory immunity in the mucosa and gut[1,10]. Als proteins, well-known as adhesins and invasins[11], not only induce intestinal adaptive immune responses but also exacerbate intestinal colitis and colon damage[12]. Among Als family proteins, Als3 is the most abundant and hypha-specific protein. It induces immune responses that promote fungal clearance during systemic infection[9].

[1]Department of Biological Chemistry, University of California, Irvine, CA, USA. [2]Division of Infectious Diseases, Lundquist Institute for Biomedical Innovation at Harbor-UCLA Medical Center, Torrance, CA, USA. [3]Department of Physiology and Biophysics, University of California, Irvine, CA, USA. [4]Division of Biology and Biological Engineering, California Institute of Technology, Pasadena, CA, USA. [5]David Geffen School of Medicine at UCLA, Los Angeles, CA, USA. [6]Present address: Gilead Sciences Inc, Foster City, CA, USA. ✉e-mail: h4liu@uci.edu

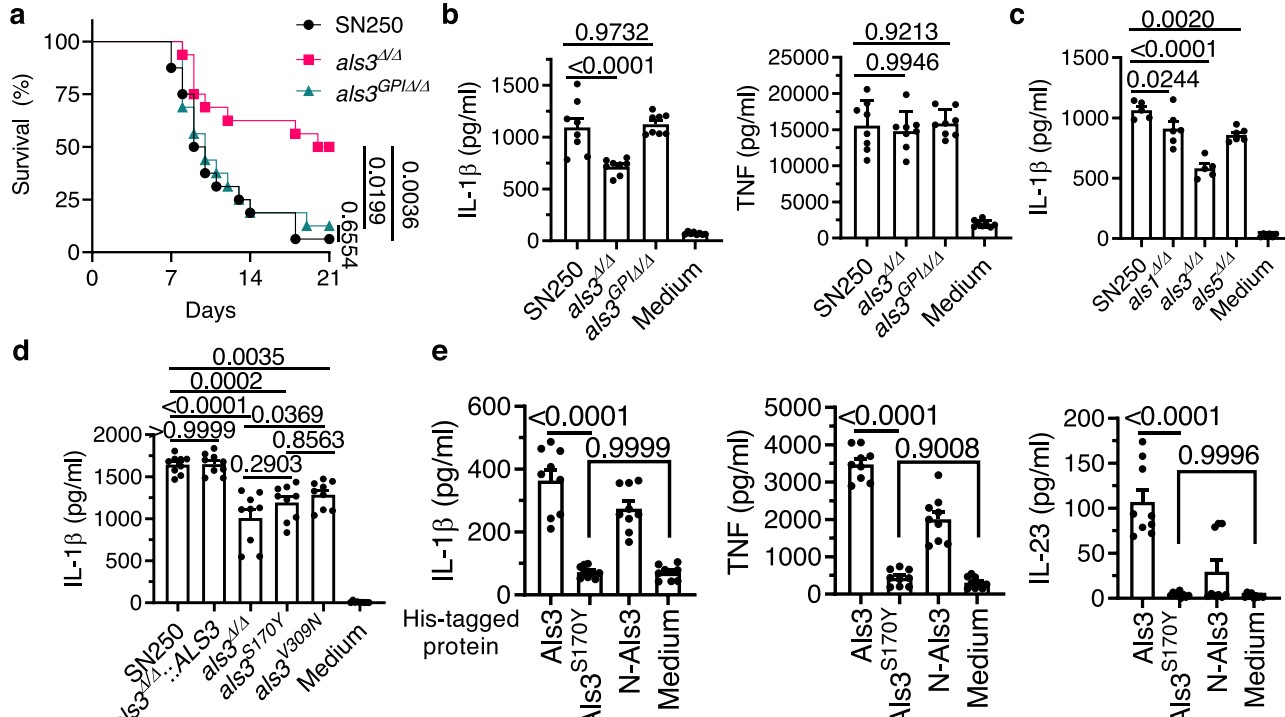

**Fig. 1 | Secreted *Candida albicans* Als3 protein induces immune responses.**
**a** Survival curves of male mice infected with $7.5 \times 10^4$ cells of WT (SN250), the $als3^{\Delta/\Delta}$ mutant, or the $als3^{GPI\Delta/\Delta}$ mutant. *P* values were calculated using a log-rank (Mantel-Cox) test ($n = 16$). The data represent the combined results of two independent experiments. ns, not significant. **b–e** ELISA analysis of secreted cytokines of BMDCs after infections with yeast cells at MOI 1 for 24 h (**b–d**) or stimulation with purified Als3 ($15 \,\mu g \, ml^{-1}$), Als3$^{S170Y}$ ($15 \,\mu g \, ml^{-1}$) and N-Als3 ($5.3 \,\mu g \, ml^{-1}$) for 24 h (**e**). The data in (**b–e**) represent the mean ± s.e.m. from three independent experiments with two or three technical repeats (**b**, $n = 8$; **d**, **e**, $n = 9$). The results in (**c**) were from two independent experiments with two or three technical repeats (SN250 and $als3^{\Delta/\Delta}$, $n = 5$; $als1^{\Delta/\Delta}$ and $als5^{\Delta/\Delta}$, $n = 6$). *P*-values were calculated using a one-way ANOVA, followed by Tukey's post-hoc test (**b–d**) or an unpaired two-tailed *t* test (**e**).

Death effector domains (DEDs), protein-protein interaction domains identified in proteins such as Fas-associated death domain (FADD), caspase-8, and cellular FLICE-inhibitory protein (cFLIP), regulate inflammation and cell death including apoptosis, pyroptosis, and necroptosis[13–15]. FADD initiates caspase-8 oligomerization via DED-DED interactions, forming a FADD/caspase-8 complex that regulates cell death pathways[16,17]. Genetic ablation of either *Caspase-8* or *Fadd* in intestinal epithelial cells (IECs) results in necroptosis and caspase-8-gasdermin-D-mediated pyroptosis-like death, leading to spontaneous ileitis and/or colitis[18–20]. Expression of catalytically inactive caspase-8$^{C362A \, or \, C262S}$ triggers the formation of caspase-8/ASC specks, activation of caspase-1, and secretion of IL-1β[14,15].

Intracellular pathogens, such as viruses, bacteria, and protozoa, must replicate without killing the host cell to survive. Therefore, their interactions with the host intracellular components have been widely studied. The cellular FLICE (caspase-8)-like inhibitory protein (short form, cFLIPs) and viral FLIPs (vFLIPs) function as apoptosis inhibitors and consist of two tandem DEDs similar to the prodomain of caspase-8[21,22]. vFLIPs inhibit death ligand-induced apoptosis by interacting with FADD and/or caspase-8 via DED-DED interaction[21].

*C. albicans* DNA packaged in extracellular vesicles triggers type I IFN signaling through the intracellular cGAS-STING pathway[23]. As an extracellular organism, *C. albicans* proteins have not been reported to directly influence intracellular cell death core components at the molecular level.

Here, we show that hyphal Als3 interacts with the DEDs of FADD and caspase-8 to modulate immune responses and cell death pathways. Als3 is a DED-interacting protein that does not have a DED but functionally resembles DED-containing proteins like vFLIPs. Although there is increasing interest in studying the recognition of fungal extracellular proteins/peptides[24,25], this intracellular recognition mechanism extends our understanding of innate immune recognition.

## Results

### Secreted Als3 protein is sufficient to stimulate inflammatory responses

Als proteins, a family of GPI-anchored cell wall proteins, function as adhesins and invasins by interacting with host cell surface proteins such as cadherins, the epidermal growth factor receptor (EGFR), and HER2[11]. GPI-anchor proteins, mainly at the cell wall, are partially released into the environment[26]. Genetically removing the GPI anchor from a protein could fully release it into the surrounding space[26]. We deleted the GPI-anchor sequence from the *ALS3* ($als3^{GPI\Delta/\Delta}$), resulting in a mutant that has lost cell surface Als3 and Als3-mediated adhesion (Supplementary Fig. 1a, b, c). The $als3^{GPI\Delta/\Delta}$ mutant secretes all Als3, whereas the WT strain secretes about 33% of its Als3 into culture supernatant (Supplementary Fig. 1a). To investigate whether the adhesion and invasion functions of Als3 determine its in vivo functionality, we infected mice with the wild-type SN250 (WT), the $als3^{\Delta/\Delta}$ mutant (full-length *als3* coding sequence knockout in WT[9]), and the $als3^{GPI\Delta/\Delta}$ mutant. The survival of mice infected with the $als3^{GPI\Delta/\Delta}$ mutant was comparable to that of mice infected with the WT strain, contrasting with the increased survival of mice infected with the $als3^{\Delta/\Delta}$ mutant (Fig. 1a). The $als3^{\Delta/\Delta}$ mutant is selectively defective in inducing IL-1β release from bone marrow-derived dendritic cells (BMDCs), a phenomenon not observed in samples infected with the $als3^{GPI\Delta/\Delta}$ variant (Fig. 1b). Therefore, Als3-mediated *C. albicans* adherence to the host cell surface or invasion into host tissues is not essential to mediate virulence. Instead, the secreted Als3 itself is sufficient to stimulate the inflammatory responses.

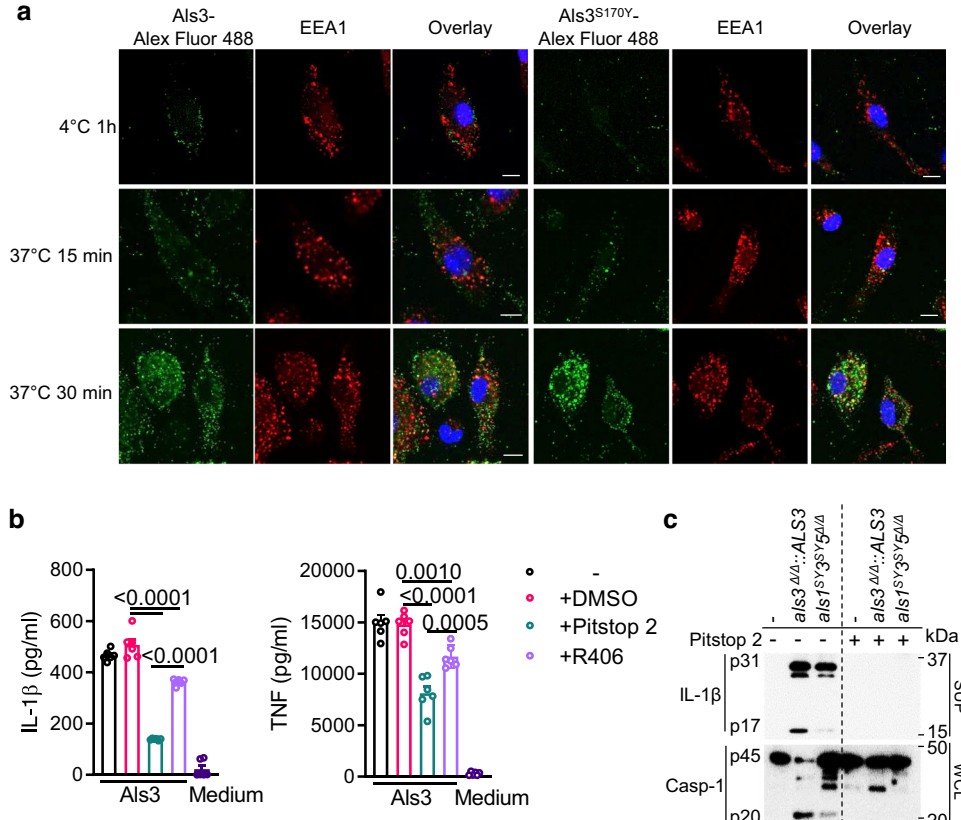

**Fig. 2 | Als3 uptake is required for inducing immune responses. a** BMDMs were incubated with Alexa Fluor 488-conjugated Als3 or Als3[S170Y] at either 4 °C (top panels) for 1 h or 37 °C for 15 or 30 min (middle and bottom panels), fixed, and then observed under confocal microscopy. Samples were stained with an antibody against the endosome marker EEA1. At least 100 cells were checked in each experiment and the pictures are representative. Scale bars, 10 μm. **b** BMDCs were stimulated with Als3 (15 μg ml⁻¹) for 24 h in the absence or presence of DMSO, Pitstop 2 (25 μM), or R406 (1 μM). IL-1β and TNF levels were measured by ELISA. Data are mean ± s.e.m. of two independent experiments. *P*-values were calculated with an unpaired two-tailed *t* test (*n* = 6). **c** BMDMs were either unstimulated (medium) or stimulated with *C. albicans* hyphae in the absence or presence of Pitstop 2. IL-1β cleavage and caspase-1 activation were analyzed by immunoblotting. Data in (**a**, **c**) are representative of three independent experiments. At least 18 randomly fields were checked for the results in (**a**, **b**).

The three most highly expressed Als proteins are Als3, Als1, and Als5. Als3, the most abundant Als protein, is hyphal-specific[27]. The *als3*[Δ/Δ] mutant exhibited the strongest defect in hyphal-induced IL-1β release by BMDCs relative to the *als1*[Δ/Δ] and *als5*[Δ/Δ] mutants (Fig. 1c). The peptide-binding cavity (PBC) and, to a lesser extent, the amyloid-forming region (AFR) of Als3 proteins play important roles in Als3-mediated adhesion[28]. We mutated Als3 S170 to Y in the PBC[9] and V309 to N in the AFR and found that both the *als3*[S170Y] mutant and the *als3*[V309N] mutant are defective in inducing IL-1β secretion by BMDCs compared to the WT strain and the *als3*[Δ/Δ]*::ALS3* complemented strain (Fig. 1d).

To further identify which Als3 residues are required to induce these immune responses, we analyzed purified fungal Als3 and its variants, including Als3[S170Y] and N-terminal Als3 (N-Als3) (Supplementary Fig. 1d). Als3 and N-Als3 stimulated BMDCs to release IL-1β, TNF, and IL-23 into the supernatant (Fig. 1e). In contrast to Als3 and N-Als3, purified Als3[S170Y] failed to stimulate BMDCs to release IL-1β, TNF, and IL-23, further confirming the vital role of the PBC in Als3-mediated immune responses (Fig. 1e).

### Als3 internalization contributes to the optimal immune responses

*C. albicans* secreted aspartyl proteases (Saps) can bind to integrins on the cell surface and be endocytosed along with these integrins[29]. We recently reported that Als3 interacts with the integrin receptor CR3 on the surface of macrophages[9]. To investigate whether Als3 can be taken up by macrophages, we compared the cell surface binding ability of Als3 and Als3[S170Y]. Alexa Fluor 488-conjugated Als3 incubated with macrophages at 4 °C for 1 h showed cell-associated fluorescence on the cell surface (Fig. 2a). In contrast, the Alexa Fluor 488-conjugated Als3[S170Y] was not detected on the surface of macrophages (Fig. 2a). After incubating at 37 °C for 15 or 30 min, conjugated Als3 and Als3[S170Y] were instead found in cytosolic organelles (Fig. 2a). Immunostaining images showed that some of the internalized Als3/Als3[S170Y] conjugates colocalized with the endosome marker EEA1 (Fig. 2a), suggesting the internalization of both conjugates from the cell surface to endosomes.

*C. albicans* hyphae are endocytosed by host epithelial cells by a clathrin-dependent mechanism where Als3 plays an important role[30]. To determine if Als3 internalization is required for Als3-mediated IL-1β release, we used an endocytosis inhibitor, Pitstop 2, which blocks endocytic ligand association with the terminal domain of clathrin and therefore, inhibits receptor-mediated endocytosis[31]. Pitstop 2 significantly reduced IL-1β release (*P* < 0.0001) and, TNF production (*P* = 0.0144) from BMDCs incubated with Als3 (Fig. 2b), indicating that receptor-mediated endocytosis of Als3 is required for cytokine production. Syk-coupled C-type lectin receptors Dectin-1 and Dectin-2 (Dectin-2 homodimer or Dectin-2/Dectin-3 heterodimer) recognize the fungal ligands β-glucan and α-mannan, respectively[9,32,33]. To investigate whether Syk activation is required for Als3-mediated immune responses, the Syk kinase inhibitor R406 was used. It only partially inhibited the release of IL-1β and TNF from BMDCs (Fig. 2b). Dectin-2

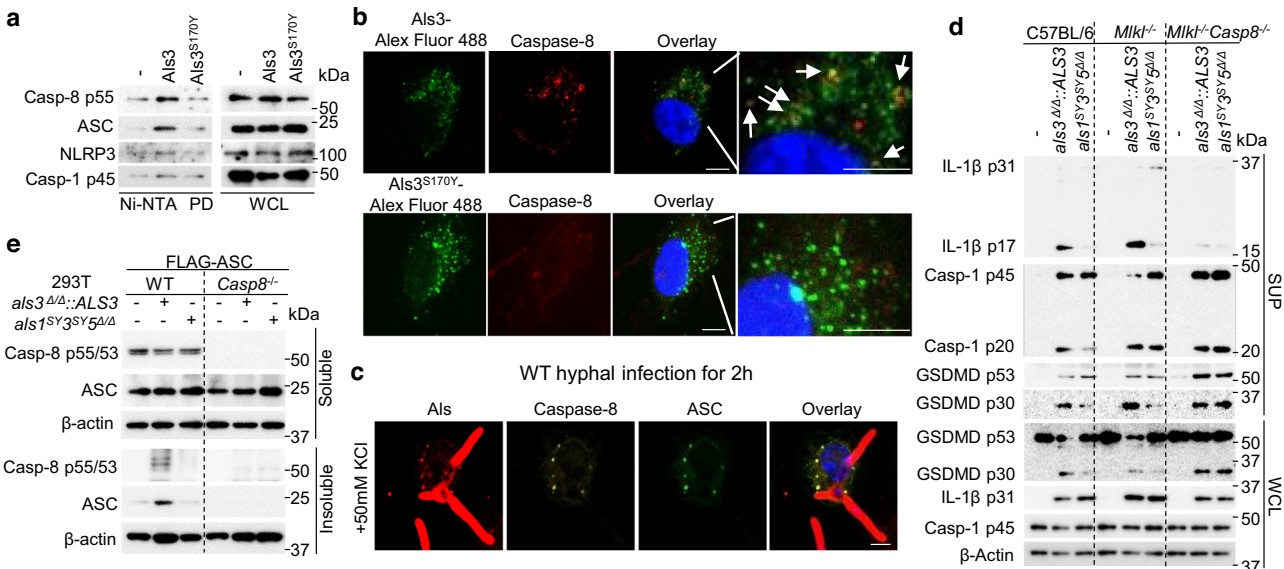

**Fig. 3 | Internalized Als3 acts through caspase-8 to regulate immune responses. a** Western blot analysis of pull-down products from LPS-primed BMDMs transfected with Als3-His or Als3[S70Y]-His for 1.5 h using Profect-P2 transfection reagent. WCL, whole cell lysates. **b** BMDMs were incubated with Alexa Fluor 488-conjugated Als3 or Als3[S170Y] at 37 °C for 1 h, fixed, and then observed under confocal microscopy. An antibody against full-length caspase-8 was used to stain the samples. Arrows at images indicate examples of colocalization of Alexa Fluor 488-conjugated Als3 with caspase-8. Scale bars, 5 μm. **c** Immunofluorescence images of BMDMs after infection with hyphae for 2 h in the presence of 50 mM KCl. Scale bars, 5 μm. Antibodies against Als, procaspase-8, and ASC were used to stain the samples. **d** Western blot analysis of the indicated markers for inflammasome activation after infection with hyphae. Results are representative of two individual experiments. **e** Western blot analysis was conducted on both WT and *Casp8*[-/-] 293 T cells. These cells were subjected to overexpression of FLAG-tagged mouse ASC (FLAG-ASC) for 14 h, followed by infection with the hyphal form of either the *als3*[Δ/Δ]*::ALS3* strain or the *als1*[SY]*3*[SY]*5*[Δ/Δ] mutant for 3 h. Results in (**a**, **d**, **e**) are representative of three independent experiments. Images in (**b**, **c**) are representative of three individual experiments with at least six randomly selected fields for each experiment.

controls Als3-mediated Syk phosphorylation[9]. The deficiency of Dectin-2 leads to a partial decrease in Als3-mediated IL-1β release (Supplementary Fig. 2), consistent with the partial role of Syk kinase in this process. The Syk inhibitor R406 had a weaker inhibitory effect compared to Pitstop 2 (Fig. 2b). Thus, Als3 internalization is more critical for signaling than activating the host Syk-dependent receptor on the cell surface. In addition, Pitstop 2 treatment completely blocked hypha-induced IL-1β processing and caspase-1 activation in BMDMs (Fig. 2c). Therefore, Als3 internalization by host cells contributes to the optimal Als3-mediated immune responses.

**Internalized Als3 acts through caspase-8 to induce ASC oligomerization**

Caspase-8 interacts with the ASC adapter and promotes ASC self-assembly during bacterial infection or sterile inflammation[14,15,34]. ASC aggregates to form the macromolecular ASC speck that serves as a platform for caspase-1 activation[35]. Als proteins have a pro-aggregation property[36], showing particle morphology on the cell surface or in cells (Fig. 2a). To explore the Syk-independent pathways involved in Als3-induced IL-1β release, we hypothesized that the internalized Als3 may directly interact with caspase-8 to promote ASC oligomerization. To expedite the cellular entry of Als3 and minimize its role in activating cell surface receptors, we used Profect P2, a protein transfection reagent, to deliver Als3 into LPS-primed BMDMs. A pull-down assay using Ni-NTA to bind Als3-His or Als3[S170Y]-His demonstrated that Als3, but not Als3[S170Y], interacted with caspase-8 and ASC in the lysates of Als3- or Als3[S170Y]-transfected BMDMs, while NLRP3 was also shown to interact with Als3 (Fig. 3a). Macrophages incubated with Alexa Fluor 488-conjugated Als3 for 1 h induced caspase-8 to form specks, and the specks colocalized with Als3 conjugates (Fig. 3b). In contrast, macrophages incubated with Als3[S170Y] conjugates did not show obvious caspase-8 speck formation (Fig. 3b). To validate the formation of a complex involving fungal Als3, caspase-8, and ASC within cells,

immunostaining was used to detect intracellular Als3 after hyphal infection. Confocal microscopy images depicted the colocalization of Als3 with caspase-8 and ASC at 2 h after *C. albicans* hyphal infection in the presence of 50 mM KCl to prevent the NLRP3 inflammasome activation (Fig. 3c and Supplementary Fig. 3a, b). These data reveal that Als3 is a fungal effector that directly interacts with a complex containing caspase-8, ASC, and NLRP3.

Caspase-8 regulates pyroptosis by inhibiting or activating its catalytic activity in distinct ways[37]. Mice lacking caspase-8 experience embryonic lethality during gestation due to uncontrolled necroptosis, a form of regulated cell death[38]. However, they survive beyond weaning when either of the necroptosis-mediating genes, *Rip3* or *Mlkl*, is co-ablated, preventing the excessive necroptotic response[38]. To determine whether Als proteins activate the inflammasome through caspase-8 in BMDMs, WT, *Mlkl*[-/-], or *Mlkl*[-/-]*Casp8*[-/-] BMDMs were infected with the *als3*[Δ/Δ]*::ALS3* strain or the *als1*[SY]*3*[SY]*5*[Δ/Δ] strain. Although pro-IL-1β levels were comparable for all infected BMDMs, the absence of caspase-8 led to a decreased level of mature IL-1β (Fig. 3d), indicating that caspase-8 is essential for hypha-mediated inflammasome activation, but not priming. We also observed comparable levels of the cleaved form of caspase-1 and Gasdermin D (GSDMD) in *Mlkl*[-/-]*Casp8*[-/-] BMDMs infected with the control strain *als3*[Δ/Δ]*::ALS3* and the *als1*[SY]*3*[SY]*5*[Δ/Δ] mutant, but not in WT or *Mlkl*[-/-] BMDMs (Fig. 3d), indicating that Als proteins mediate inflammasome activation through caspase-8.

To mechanistically characterize the ability of Als3 to promote caspase-8-mediated ASC speck formation, ASC was transiently transfected into both WT human embryonic kidney 293 T cells and *Casp8*[-/-] 293 T cells. ASC was increasingly detected in the Triton X-100 insoluble cellular fraction of WT cells infected with the *als3*[Δ/Δ]*::ALS3* strain but not the *als1*[SY]*3*[SY]*5*[Δ/Δ] mutant (Fig. 3e). In *Casp8*[-/-] 293 T cells infected with the *als3*[Δ/Δ]*::ALS3* strain, no ASC aggregates were detected (Fig. 3e). Disuccinimidyl suberate (DSS)-crosslinking of the insoluble cellular fractions confirmed that more ASC dimers formed after infection with the

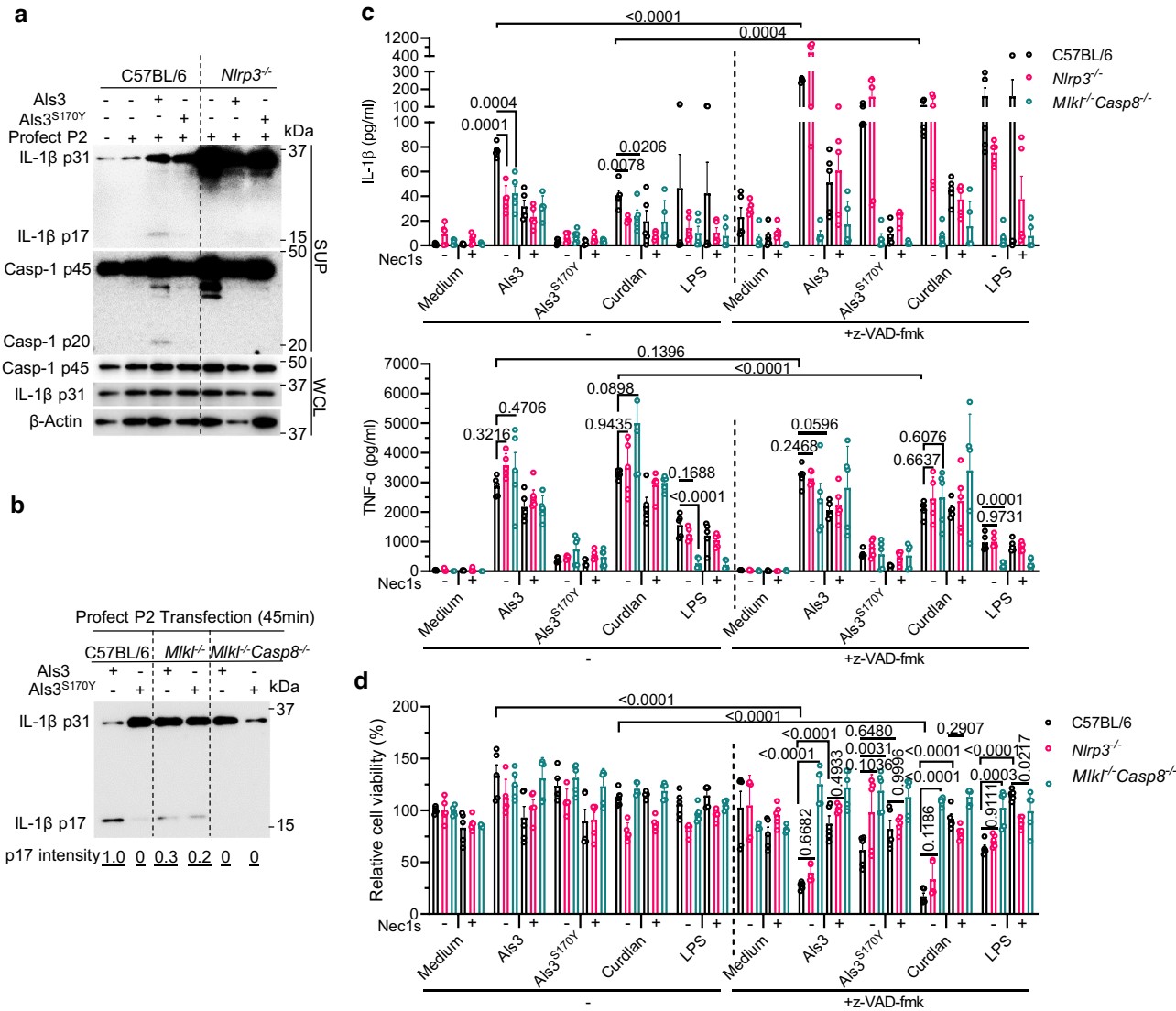

**Fig. 4 | Als3-mediated IL-1β processing requires NLRP3, MLKL, and caspase-8. a**, **b** LPS-primed BMDMs were untransfected or transfected with Als3-His or Als3^S170Y-His (5 μg ml⁻¹), and caspase-1 (p10) and/or IL-1β (p17) processing was measured by immunoblotting. Profect-P2 transfection reagent was used for protein transfection. BMDMs were primed with 200 ng ml⁻¹ LPS for 3 h. BMDMs from mouse mutants including *Nlrp3*⁻/⁻ (**a**), *Mlkl*⁻/⁻ (**b**), or *Mlkl*⁻/⁻*Casp8*⁻/⁻ (**b**), were used together with WT. **c**, **d** GM-CSF cultured BMDMs were stimulated with the indicated agents for 2 days or LPS for 3 h. The levels of IL-1β and TNF in supernatants were quantitated by ELISA. Als3/Als3^S170Y, 15 μg ml⁻¹; Curdlan, 1 mg ml⁻¹; LPS, 200 ng ml⁻¹,

Nec1s 10 μM, z-VAD-fmk 50 μM. Cell viability was normalized based on the viability of cells cultured with medium only. Data in (**c**, **d**) represent the mean ± s.e.m. of data from two independent experiments with two or three technical repeats (*n* = 5). *P*-values for the samples within the group with or without z-VAD-fmk were calculated with one-way ANOVA with Tukey post-hoc analysis. *P*-values for the samples between the groups with or without z-VAD-fmk were calculated with an unpaired two-tailed *t* test. The data presented in (**a**, **b**) are representative of three independent experiments.

*als3*^Δ/Δ::*ALS3* strain as compared to the infection with the *als1*^SY*3*^SY*5*^Δ/Δ mutant (Supplementary Fig. 3c). Collectively, these findings demonstrate that Als3 induces ASC oligomerization through caspase-8.

**The NLRP3 inflammasome, MLKL, and caspase-8 are required for Als3-induced IL-1β processing**
To dissect the pathways responsible for Als3-induced IL-1β processing, mouse mutants and/or inhibitors were used. Immunoblotting confirmed that the transfected Als3 activated the NLRP3 inflammasome in macrophages within 45 min, as indicated by the cleaved forms of caspase-1 and IL-1β in the supernatant of Als3-transfected wild-type BMDMs, but not *Nlrp3*⁻/⁻ BMDMs (Fig. 4a). In contrast, the transfected Als3^S170Y did not induce NLRP3 inflammasome activation (Fig. 4a). Compared to wild-type BMDMs, the transfected Als3 in *Mlkl*⁻/⁻ BMDMs showed a reduced level of IL-1β processing (Fig. 4b). These results

demonstrate that necroptotic signaling promotes Als3-mediated IL-1β processing, in line with downstream necroptotic MLKL oligomerization and membrane permeabilization, which is associated with potassium ion efflux and the activation of NLRP3 inflammasome[39]. Unlike the results from immunoblotting with BMDMs infected hyphae (Fig. 3d), MLKL played an important role in the purified Als3-induced IL-1β release (Fig. 4b), suggesting that other hyphal ligand-mediated pathways also modulate the balance between inflammasome signaling and necroptotic signaling in BMDMs. In *Mlkl*⁻/⁻*Casp8*⁻/⁻ BMDMs, Als3-induced IL-1β processing was completely abrogated (Fig. 4b). Notably, the absence of caspase-8 did not affect Als3-induced IL-1β production (Fig. 4b), which is distinct from the reported role for caspase-8 in the LPS- or bacteria-induced inflammasome priming step[40]. Collectively, these data indicate that NLRP3, MLKL, and caspase-8 are critical for Als3-induced IL-1β processing.

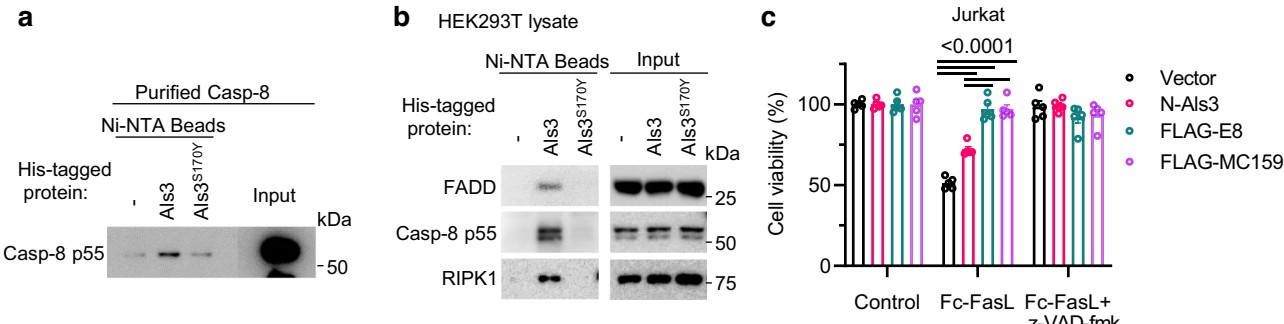

**Fig. 5 | Als3 inhibits apoptosis. a** Purified Als3-His directly interacts with recombinant human full-length caspase-8. Pull-down of human full-length caspase-8 with N-Als3-His or N-Als3$^{S170Y}$-His. Results are representative of three individual experiments. **b** Pull-down of Als3-interacting proteins in 293 T lysates using Ni-NTA magnetic beads. Results are representative of three individual experiments. **c** The susceptibility of Vector, N-Als3, FLAG-E8, or FLAG-MC159 expressing Jurkat cells to apoptosis was determined by incubating cells with the indicated concentration of Fc-FasL (ng ml⁻¹) for 24 h. Data are mean ± s.e.m. from two independent experiments with two or three technical repeats ($n = 5$). P-values were calculated using two-way ANOVA with Tukey's post-hoc test.

The kinase activity of RIPK1 is essential for necroptotic signaling. Necrostatin-1 (Nec1) and its improved analog Nec1s interact with RIPK1 at the back pocket of the ATP-binding site to block RIPK1 kinase activity directly[41,42]. Treating BMDCs with either Nec1s or the NLRP3 inhibitor MCC950 partially decreased Als3-induced IL-1β release, and the combination of both inhibitors further inhibited the release of this cytokine (Supplementary Fig. 4). Similarly, the absence of NLRP3 or MLKL together with caspase-8 led to a decrease in Als3-induced IL-1β release in BMDMs (Fig. 4c). Taken together, our data indicate that Als3-mediated IL-1β release depends on the NLRP3 inflammasome and necroptotic signaling.

Of note, the purified Als3 protein did not kill the macrophages (Fig. 4d), indicating that the pores formed to release IL-1β were either repaired or not sufficiently numerous to induce cell death. Curdlan served as a positive control for the requirement of caspase-8 in IL-1β processing (Fig. 4c). This result is consistent with the report that curdlan-induced IL-1β processing requires caspase-8 in a Dectin-1/Syk pathway-dependent manner[43]. We also noticed that curdlan did not induce macrophage cell death in our assay, aligning with prior reports that curdlan does not consistently induce cell death in BMDMs[44–46]. In addition, curdlan has been reported to induce granulocyte-macrophage colony-stimulating factor (GM-CSF) production directly and, therefore, enhance cell proliferation in resident macrophages[47]. We observed that the treated BMDMs had better proliferation than the medium-alone control, indicating that this GM-CSF autocrine mechanism might be activated and overcome cell death.

*C. albicans* can activate necroptotic signaling in BMDMs, which is further strengthened by pan-caspase inhibition with z-VAD-fmk[48]. We found that the presence of z-VAD-fmk significantly increased ligand-induced BMDM cell death, especially Als3- and curdlan-induced cell death, consequently releasing more of the cell-death-dependent cytokine, IL-1β (Fig. 4c, d). By contrast, TNF levels did not show the same trend (Fig. 4c). We observed that the amount of TNF induced by LPS in *Mlkl$^{-/-}$Casp8$^{-/-}$* BMDMs was significantly lower than that in wild-type BMDMs (Fig. 4c). This reduction was not observed when *Mlkl$^{-/-}$Casp8$^{-/-}$* BMDMs were stimulated with Als3 or curdlan. These results are consistent with the finding that caspase-8 is not required for Als3- or curdlan-mediated priming (Fig. 4b)[44].

**Als3 forms complexes with core cell death pathway components, including RIPK1, FADD, and caspase-8, and inhibits apoptosis**

To determine whether Als3 directly interacts with caspase-8, His-tagged Als3 and recombinant human caspase-8 were used to analyze potential protein-protein interactions. Pull-down assays confirmed a specific interaction between purified Als3 and caspase-8, whereas Als3$^{S170Y}$ did not interact with caspase-8 (Fig. 5a).

In cell death signaling platforms, FADD and caspase-8 are not only key proteins of the death-inducting signaling complex (DISC) but are also core components of the cytosolic TNF complex II[49], the ripoptosome[50,51], and the RIPK1/RIP3-containing necrosome[52]. To identify Als3-associated proteins, we used Als3-His-coated beads to perform pull-down experiments in 293 T lysates. We found that RIPK1, FADD, and caspase-8 could be specifically pulled down by Als3 (Fig. 5b), indicating that Als3 is associated with RIPK1/FADD/caspase-8.

*C. albicans* infection inhibits apoptosis of human monocytes, monocytic U937 cells, and Caco-2 cells[53,54]. Given that Als3 and vFLIPs both bind DED(s), we investigated the role of Als3 in apoptosis. Stable expression of N-Als3 in Jurkat cells partially inhibited apoptosis triggered by Fas ligand (FasL), and apoptosis could be fully blocked by the pan-caspase inhibitor z-VAD-fmk (Fig. 5c). N-Als3 is weaker at inhibiting apoptosis relative to the vFLIPs, E8 and MC159[21] (Fig. 5c).

**Als3 promotes caspase-8 and FADD oligomerization through interaction with their DED(s)**

Caspase-8 can induce the production of cytokines and ASC-caspase-1 activation by serving as a scaffold protein dependent on its prodomain, the tandem DEDs[14,15,55]. Caspase-8 DEDs form filaments in vitro[16]. The DED in FADD also mediates the formation of FADD oligomers for stable interaction with an activated death receptor[56], which in turn nucleates caspase-8 oligomerization to mediate cell death[17]. Transfection of *Casp8$^{-/-}$* 293 T cells with EGFP-N-Als3, along with FLAG-caspase-8$^{C360A}$ (a catalytically inactive caspase-8) or tagBFP-FLAG-FADD, resulted in an increased aggregation/filament formation of caspase-8$^{C360A}$ or FADD that was present in the Triton X-100 insoluble fractions (Fig. 6a, b). Notably, mutations in the PBC (N-Als$^{S170Y}$), the AFR N-(Als3$^{V309N}$), or in both the PBC and AFR (N-Als$^{S170YV309N}$) resulted in compromised caspase-8$^{C360A}$ or FADD oligomerization compared to WT N-Als3 (Fig. 6a, b). Therefore, both the PBC and AFR of Als3 are required for promoting the oligomerization of caspase-8 or FADD.

Caspase-8 variants with mutations in DEDs, F122E (type I interface) and K148D/R149E (type II interface), are defective in filament formations in vitro and whole cells[16]. The co-expression of N-Als3 with these variants failed to rescue this defect (Fig. 6c). Immunoprecipitation with the soluble fractions of the transfected *Casp8$^{-/-}$* HEK293T cell lysates indicated that F122, but not K148/R149, is critical for N-Als3 binding to caspase-8 DEDs (Fig. 6c). Furthermore, fluorescence microscopy images of cells transfected with EGFP-N-Als3 and DsRed-caspase-8$^{C360A}$

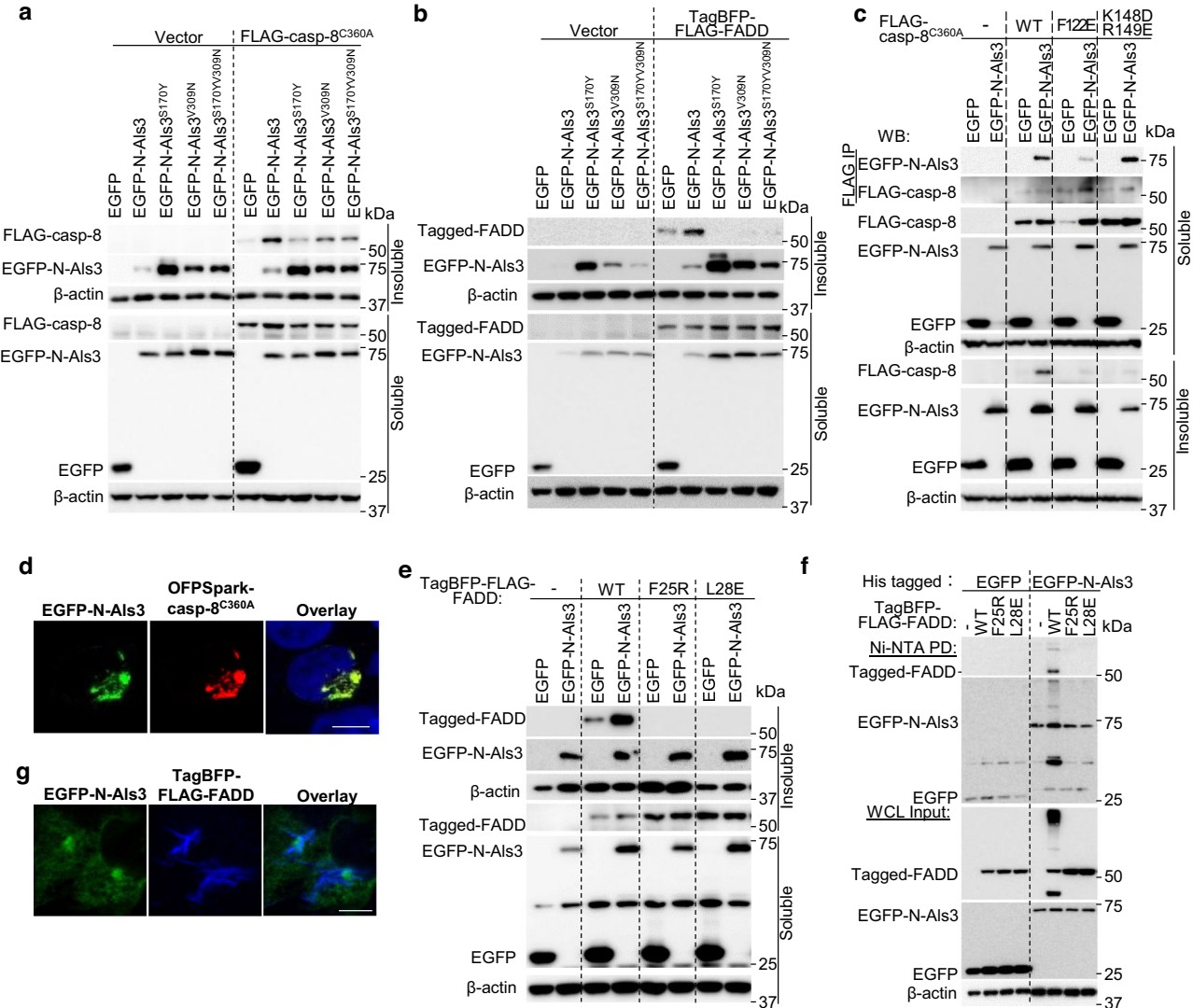

**Fig. 6 | Als3 promotes the oligomerization of catalytically inactive caspase-8 and FADD by interacting with their DED(s). a, b** Western blot analysis of *Casp8*[-/-] 293 T cells transfected with EGFP or EGFP-N-Als3/variants and FLAG-caspase-8[C360A] or tagBFP-FLAG-FADD. **c** Immunoprecipitation and Western blot analysis of *Casp8*[-/-] 293 T lysates overexpressing either empty vector (−), FLAG-tagged human caspase-8[C360A], caspase-8[C360AF122E], or caspase-8[C360AF122EK148DR149E] with EGFP or EGFP-N-Als3. **d** Confocal fluorescence images of *Casp8*[-/-] 293 T cells overexpressing human caspase-8[C360A] and EGFP-N-Als3. The blue staining was from DAPI. Scale bar, 10 μm.

**e, f** Immunoprecipitation and Western blot analysis of *Casp8*[-/-] 293 T lysates overexpressing either empty vector (−), tagBFP-FLAG-tagged human FADD, FADD[F25R], or FADD[L28E] with EGFP-N-Als3. The insoluble fraction of (**e**) was dissolved in a buffer with urea as in all other insoluble fraction analyses, which was not applied for immunoprecipitation in (**f**). **g** Confocal immunofluorescence images of *Casp8*[-/-] 293 T cells overexpressing human FADD and EGFP-N-Als3. Scale bar, 5 μm. All results are representative of at least three individual experiments. At least 18 randomly fields were checked for the results in (**d, g**).

revealed that caspase-8[C360A] co-localized with N-Als3 aggregates (Fig. 6d). Therefore, N-Als3 interacts with caspase-8 DEDs to promote their aggregation.

To identify the interface of FADD/N-Als3 association, we generated and examined several FADD DED point mutants. Both the FADD F25R and L28E mutations are defective in self-association[56]. They also failed to form Triton-X100 insoluble complexes even with N-Als3 co-expression (Fig. 6e). A His-tag pull-down of soluble fractions of the transfected *Casp8*[-/-] HEK293T cell lysates showed that FADD F25R and L28E are defective in interacting with N-Als3 (Fig. 6f). A68D, which is defective for CD95 binding but has only modest defects in FADD self-association[56], showed a moderate defect in FADD/N-Als3 association (Supplementary Fig. 5a). D44R, which is not required for self-association of FADD[56], showed no defect in binding N-Als3 (Supplementary Fig. 5a). These results suggest that N-Als3 interacts with FADD via the same interface of DED that is used for self-association and

oligomerization. Moreover, fluorescence microscopy images of cells transfected with tagBFP-FLAG-FADD and EGFP-N-Als3 revealed that FADD fibers extended from Als3 aggregates, with Als3 residing in the center when the complex had more than one branch (Fig. 6g). This was strikingly different from the colocalization of caspase-8[C360A] and N-Als3 (Fig. 6d). Together, our data strongly indicate that Als3 promotes FADD aggregation.

Overexpression of two tandem DEDs of caspase-8 (DED1/2) leads to DED filamentation, while overexpression of the first DED (DED1) alone does not[57]. DED1/2 and DED1 of human caspase-8 were transfected into *Casp8*[-/-] HEK293T cells to assess their ability to promote caspase-8[C360A] or FADD oligomerization. N-Als3 and DED1/2, but not N-Als3[S70Y] and DED1, elevated caspase-8[C360A] and FADD in Triton-X100 insoluble fractions (Supplementary Fig. 5b). Collectively, our data suggest that N-Als3 mimics DED1/2 of caspase-8 in promoting DED oligomerization.

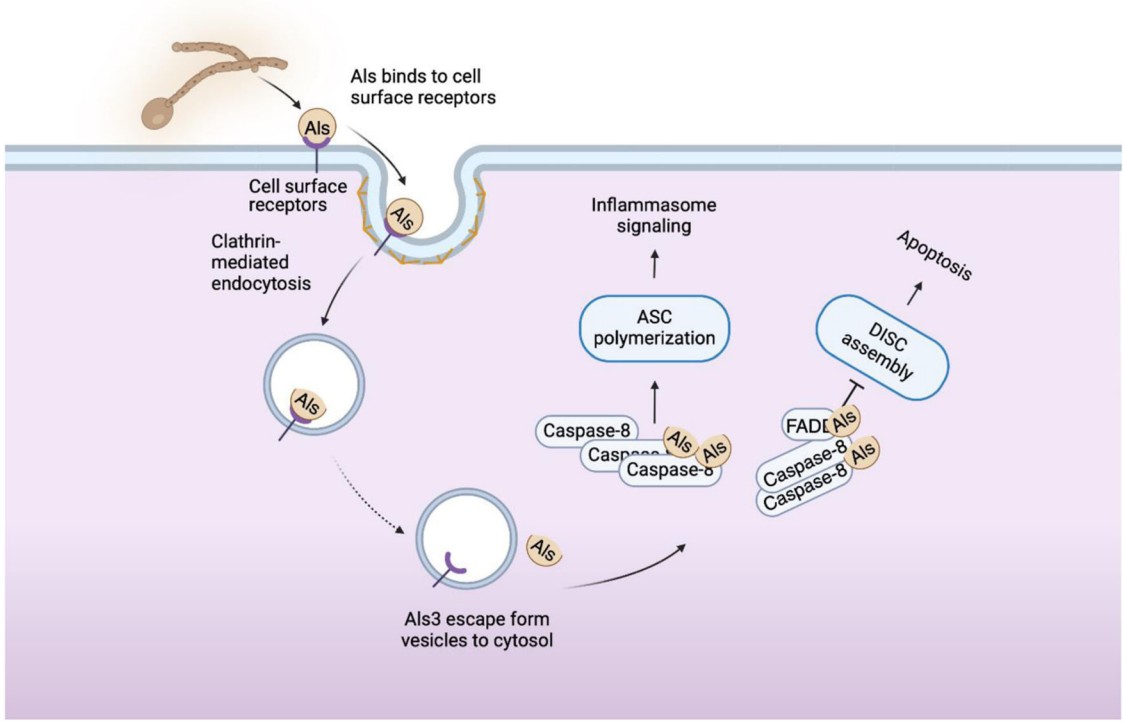

**Fig. 7 | Internalized Als3 modulates inflammasome signaling and apoptosis.** Als proteins from *C. albicans* hyphae are internalized into host cells. The specific mechanisms by which Als proteins escape from endosomes remain unknown. Within the cytosol, Als proteins interact with the DEDs of FADD and/or caspase-8, promoting their oligomerization and forming an N-Als/FADD/caspase-8 complex. This complex plays multiple roles in different cell types. It promotes ASC oligomerization in macrophages and inhibits apoptosis in non-immune cells. Consequently, Als not only inhibits apoptosis but also promotes inflammasome signaling. Created in BioRender. Zhou, T. (2025) https://BioRender.com/u12i251.

## Discussion

Als proteins serve as adhesins and invasins by binding to cadherins/EGFR/integrins, thereby inducing endocytosis by host cells in vitro[9,11], which contributes to virulence during oropharyngeal candidiasis. Here, we showed that deletion of the Als3 GPI anchor does not impair *C. albicans* virulence in the mouse model of hematogenously disseminated candidiasis. Instead, purified Als3 induces immune cells to release cytokines, which we showed is partially depend on CR3 on the surface of immune cells[9]. In the current study, we demonstrated that Als3 also acts intracellularly and binds with the DEDs of core cell death components FADD and caspase-8. DEDs of caspase-8 and FADD bridge Als3 to ASC for inflammasome activation and connect Als3 to apoptosis inhibition (Fig. 7). This study, therefore, shows an example of fungal proteins targeting core components of the mammalian intracellular cell death pathway. Future studies will examine other members of the Als family proteins.

Host cells recognize pathogens through cell surface and/or intracellular receptors to activate immune responses and mediate microbial clearance. *C. albicans* β-glucan can be recognized by immune cells via their cell surface receptors, including Dectin-1 and CR3. β-glucan on heat-killed *C. albicans* and in curdlan triggers Syk-dependent formation of the CARD9–Bcl-10–MALT1 scaffold that results in IL-1β transcription as well as the formation and activation of a MALT1–caspase-8–ASC complex that mediated the processing of pro-IL-1β[43]. Dectin-1 deficiency or Syk inhibition is sufficient to completely block IL-1β release from BMDCs stimulated with heat-killed *C. albicans*[43], whereas live *C. albicans* partially rely on CR3 but not Dectin-1 for IL-1β release[44]. In comparison, Syk is only partially involved in Als3-induced IL-1β release. Also, caspase-8 deficiency completely abrogated this release, suggesting Syk is not essential for the Als3/caspase-8 axis. Here, we showed that Als3 directly interacts with caspase-8 and forms a complex at the molecular level through DEDs. Thus, host cells that do not express Syk-coupled receptors can also sense hypha by their DED-containing protein caspase-8 and FADD. Although Als3/caspase-8 was discovered in immune cells, it might also play an important role in regulating inflammation in non-immune cells that express caspase-8.

The deficiency of Dectin-2 led to a decrease in N-Als3-induced IL-1β release in BMDCs (Supplementary Fig. 2), indicating that the mannans of N-Als3 contribute to its activity. However, mutation of the PBC completely abolished Als3's activity in inducing immune responses, although the mutant protein Als3^S170Y still retains glycosylation. This suggests that protein structure determines whether Als3 is active, while glycosylation contributes to the level of activity.

Fungi, including both yeast and filamentous forms, are increasingly recognized for their production of extracellular vesicles (EVs) containing a wealth of proteins, lipids, and nucleic acids. During host-pathogen interactions, EVs play an important role in delivering pathogen components to the host. Bacterial vesicles are responsible for LPS entering host cells[58]. The EVs from hyphae, but not yeast, have a cytotoxic effect on human macrophages and can elicit TNF production[59]. A significant amount of Als3 is secreted during in vitro growth, and Als3 was identified as an abundant protein in hyphal EVs[59]. Thus, EVs may be involved in enhancing the delivery of Als3 into host cells. Although the exact mechanism is unknown, it is also possible that after *C. albicans* hyphae are phagocytosed, Als proteins may escape from the phagosome and enter the cytoplasm, aided by other hyphal proteins and/or progressive hyphal elongation.

The current study demonstrates the association of fungal Als3 with DED(s) of caspase-8 and FADD that regulates ASC speck assembly

and cell death. We showed that Als3 enhances the formation of FADD and caspase-8 filaments. Importantly, FADD is present at sub-stoichiometric levels compared to caspase-8[60]. Thus, Als3 might hijack FADD more efficiently than caspase-8. Cryo-EM structural analysis showed that caspase-8 filament formation is nucleated by FADD via DED interactions[17], as evidenced by the sequential assembly of FADD short filaments followed by caspase-8 long filaments. Since N-Als3 completely colocalizes with caspase-8, we speculate that it might share more features with the DEDs of caspase-8 compared to the DED of FADD.

FADD and caspase-8 are core components of cell death complexes, including the DISC, ripoptosome, and necrosome. Many γ-herpesviruses, including the HHV8, herpesvirus saimiri, equine herpesvirus 2, bovine herpesvirus 4, and moloscum contagiosum virus, encode vFLIP, which disrupt recruitment of procaspase-8 to the DISC[61,62]. Like vFLIPs, Als3 interacts with FADD or caspase-8, inhibiting apoptosis induced by the activation of death receptors. Moreover, Als3 promotes inflammasome and necroptotic signaling by facilitating caspase-8/ASC and RIPK1/FADD/caspase-8 assembly. The hijacking of FADD and caspase-8 by Als3 aligns with exacerbated colitis caused by Als proteins[12] or the genetic ablation of either caspase-8 or FADD in IECs[12,18–20].

Lastly, the increased levels of *C. albicans* in tumor microenvironments have been linked to poor prognoses in cancer patients[63]. *C. albicans* stimulation of EGFR and c-Met likely enhances cancer progression. Moreover, the loss of apoptotic signaling is a hallmark of cancer[64]. Compared to vFLIPs, N-Als3 has a less pronounced effect on inhibiting death-ligand-induced apoptosis. *C. albicans* exists as a commensal in most individuals, and therefore Als proteins might impact host cell death. Some viral and bacterial infections also inhibit host cell apoptosis[65]. Notably, the Kaposi sarcoma herpesvirus vFLIP, which possesses two DED domains and shares structural similarities with cFLIPs, induces tumorigenesis in mice[66]. In addition to Als3-exaggerated inflammation, it is tempting to speculate that Als proteins may contribute to the cancer susceptibility of patients with increased *C. albicans* colonization.

## Methods

### Ethics
All studies described have been approved by UC Irvine oversight committees, including the use of mice as a source of macrophages and dendritic cells by the UC Irvine IACUC. All systemic infection work was approved by the Institutional Animal Care and Use Committee at the Lundquist Institute for Biomedical Innovation at Harbor-UCLA Medical Center.

### Mice
C57BL/6 J and *Nlrp3*[-/-] (C57BL/6 J background) mice were purchased from Jackson Laboratories (Bar Harbor, ME, USA). Dectin-2-deficient mice (C57BL/6 J background) were a kind gift from Yoichiro Iwakura (University of Tokyo, Tokyo, Japan). 6-week-old male and female C57BL/6 J mice were purchased for in vivo experiments. All animals were bred in pathogen-free conditions in micro isolator cages and were treated according to institutional guidelines following approval by the University of California IACUC. Mice were kept between $22.2 °C \pm 1.1 °C$ with a humidity of 50−55% and 12 h lights on/12 h lights off cycle. $CO_2$ was used to euthanize mice. Bones from *Mlkl*[-/-] (C57BL/6 background) and *Mlkl*[-/-] *Casp8*[-/-] mice (C57BL/6 background) were from Dr. Egil Lien at UMASS Chan Medical School, Wooster, MA, and described previously[67]. We routinely use male and female mice aged 6-10 weeks for in vitro assays.

### Culture of *C. albicans*
Single colonies of *C. albicans* strains from yeast-peptone-dextrose (YPD) agar plates were inoculated into YPD liquid media, followed by overnight culture at 30 °C (yeast forms). For the preparation of hyphae, yeast cells were cultured in synthetic complete (SC) media with 2% N-acetylglucosamine (GlcNAc) as the sole carbon source at 37 °C in cell culture plates.

### *C. albicans* strains
*C. albicans* SN250 (ARG4[+]) was used as a WT strain. Gene editing by CRISPR/Cas9 technology optimized for *C. albicans* was used as previously described[68] to generate the *als3*[GPIΔ/Δ], *als3*[V309N], and *als3*[C-Als3Δ/Δ]::7x His (N-Als3 17-432) mutants. Oligonucleotides were listed in Supplementary Table 1. A restriction site was introduced into the point mutation site, which was used to identify the correct mutation after PCR. In addition to digestion, the PCR product was further confirmed by sequencing.

### Production and purification of Als3-His, Als3[S170Y]-His and N-Als3-His
Als3-His, Als3[S170Y]-His, N-Als3-His (17-432) and N-Als3[S170Y]-His (17-432) were purified from the supernatant of *als3*[GPIΔ/Δ]::7x His, *als3*[S170Y GPIΔ/Δ]::7x His, *als3*[C-Als3Δ/Δ]::7x His mutants, and *als3*[S170Y C-Als3Δ/Δ]::7x His mutants respectively, according to a previous protocol[9]. The purity of each protein was verified by SDS-PAGE, showing one single band.

### Generation of recombinant caspase-8 protein
Full-length recombinant caspase-8 was produced and purified from *E. coli* ER2566 cells using the Impact Intein purification system (New England BioLabs Inc.) according to the manufacturer's instructions.

### Cell adhesion assay
These assays were conducted in a twelve-well plate format. HK-2 cells were purchased from ATCC (CRL-2190) and cultured in DMEM/F12 (Gibco) supplemented with 10% FBS (Corning), and 1% penicillin-streptomycin (Gibco). Cells formed confluent monolayers in a twelve-well tissue culture-treated polystyrene plate, approximately $5 \times 10^5$ cells. The growth medium was changed with a fresh complete medium 1 day prior to the adhesion assay. *C. albicans* strains were inoculated from the stock plate into 10 ml liquid YPD and grown overnight at 30 °C. Cells were washed once with PBS and counted. 330 *C. albicans* cells in RPMI medium were inoculated into each well and cultured for 3 h. To remove unattached *C. albicans* cells, the plate was washed once with PBS. The well was covered with 1 ml YPD top agar at 44 °C. The cultures were incubated overnight at 30 °C and c.f.u. were counted.

### Generation of BMDMs and BMDCs
BMDMs were derived by culturing bone marrow cells in DMEM medium (Gibco) containing 10% FBS (Corning), 1x penicillin-streptomycin (Gibco) (referred to as complete DMEM medium) with 20 ng mL$^{-1}$ M-CSF (R&D). Medium with fresh M-CSF was added every other day. At day 5 or 6, the medium was replaced with the complete DMEM medium with 20 ng ml$^{-1}$ murine GM-CSF (Peprotech) and further cultured for 24−36 h. Alternatively, bone marrow cells were directly cultured with RPMI 1640 (Gibco) supplemented with 10% FBS, 1% penicillin-streptomycin, and 20 ng mL-1 GM-CSF (Peprotech) for 7 days. Culture medium was replaced every other day. The adherent cells were harvested.

BMDCs were derived by culturing bone marrow cells with RPMI 1640 medium as above for 7 days. Culture medium was replaced every other day. On day 7, the cells in suspension and the semi-adherent cells lifted by gently washing with PBS were harvested.

### Infection or stimulation of BMDMs and BMDCs
To measure cytokine release, $1 \times 10^5$ *C. albicans*/well were seeded into 96-well plates and cultured at 37 °C. BMDMs or BMDCs were added to the plate at MOI 1. If not indicated, BMDCs were infected with the yeast form of *C. albicans* for 24 h, BMDMs were infected with hyphae for

5.5 h, and BMDCs were challenged with purified proteins for 24 h. The cultures were centrifuged at $250 \times g$ for 2 min before incubation. For Western blot analysis, the infections were scaled up into 48-well plates.

## In vivo Systemic *Candida* infections

In vivo, animal work was approved by the Institutional Animal Care and Use Committee of the Los Angeles Biomedical Research Institute. *C. albicans* strains were cultured in YPD broth and grown at 30 °C, 200 rpm. Yeast cells were washed in PBS, enumerated, and injected into male C57BL/6 mice via the lateral tail vein. Animals were randomly assigned to the different groups. Researchers were not blinded to the experimental groups because the endpoint survival was an objective measure of disease severity. For survival experiments, mice were monitored twice daily, and moribund mice were humanely euthanized. Both male and female mice were used in the systemic infection experiments. However, there was no significant difference in survival rates between females infected with the WT strain and those infected with the *als3*[Δ/Δ] mutant. As a result, the survival data for females was not included in the analysis.

## Released Als3 detection

Overnight *Candida albicans* cultures grown in YPD broth at 30 °C were washed, counted, and adjusted to match the lowest culture density. These washed cultures were then diluted 1:100 in 5 mL of SC medium containing 2% GlcNAc and incubated at 37 °C for 24 h. After incubation, the supernatant was collected and concentrated using 4 mL Amicon Ultra centrifugal filters (100 kDa, UFC8100). One-tenth of the final volume was loaded onto a 7% SDS-PAGE gel. Detection of secreted Als3 was performed by Western blotting using an anti-Als antibody derived from rabbits immunized with an Als3 fragment (described previously[69], 1: 250).

## Expression vectors

N-Als3 and truncations with codon optimized for human cell expression were cloned into pEGFP-C1 (Clontech) with a linker of 10x His. N-Als3, FLAG-E8, FLAG-MC159 (MC159 is a kind gift from Dr. Michael Lenardo at NIH) were also cloned into the lentiviral vector pLV-EF1a-IRES-Puro (Addgene, Plasmid # 85132). Full-length human caspase-8 and truncations together with N-terminal OFPSpark, a DsRed derivative from pCMV3-C-OFPSpark® Vector (SinoBiologocal), were cloned into pCDNA3 (Invitrogen). Human caspase-8 and truncations were also cloned into pCMV-3Tag-6 (Stratagene) to have 3x FLAG at the N-terminal. FLAG-tagged full-length human FADD and truncations were cloned into mTagBFP2-Lifeact-7 (Addgene) to replace Lifeact. Plasmid pcDNA3-N-Flag-mASC1 was from Addgene.

## Cell cultures

All the cell lines have been tested to be mycoplasma negative by the commonly used PCR method but not authenticated. *Casp8*[-/-] 293 T cells and their parental 293 T cells were kindly provided by H. Kashkar Fitzgerald (University of Cologne, Cologne, Germany). Jurkat Clone E6-1 was from ATCC (TIB-152). 293 T cells were grown in DMEM. Jurkat cells were grown in RPMI. All media were supplemented with 10% (vol/vol) fetal bovine serum (FBS), 100 μml[-1], and 100 μg ml[-1] penicillin-streptomycin. All cells were grown at 37 °C in a 5% CO$_2$ incubator. 293 T cells were transiently transfected for 16 h with FuGENE HD (Promega) for *C. albicans* infection experiments and with polyethylenimine (Polysciences) for other experiments by following the manufacturers' instructions.

## Stable cell-line construction

Stable expression of N-Als3, FLAG-E8, or FLAG-MC159 in Jurkat cells was achieved via the transduction of the lentiviral vector pLV-EF1a-IRES-Puro with the infused target genes. To produce viral particles, $2 \times 10^6$ Lenti-X 293 T cells were plated on 10 cm plates and grown in DMEM/F12, 10% FBS, and 1x penicillin-streptomycin. The medium was then replaced with 5 mL of serum-free DMEM/F12 per plate. Five μg of lentiviral plasmid, 2.5 μg pMD2.G plasmid (Addgene, Plasmid # 12259), and 2.5 μg psPAX2 plasmid (Addgene, Plasmid # 12260) were co-transfected with Lipofectamine 2000 transfection reagent (Invitrogen, 11668019) into the Lenti-X 293 T cells according to manufacturer's protocol. Four hours later, the medium was changed to 10 mL of complete medium, and the cells were left to generate viruses for 3 days. Next, the medium was harvested, spun down at $500 \times g$ for 5 min, and passed through a 0.45 μm filter. The virus-containing medium was then added to a six-well plate containing the cell type of interest at a confluency of approximately 60%. The cells were left in the viral medium for 3 days, and then 2 μg ml[-1] puromycin was added to select cells for 3 days.

## Cytokine ELISA

Cytokine levels were determined using mouse IL-1β, TNF, and IL-23 uncoated ELISA kits (Invitrogen, #88-7013A-88 for IL-1β, # 88-7324-88 for TNF, and # 88-7230-88 for IL-23) according to the manufacturer's instructions. For ELISA with BMDCs, which produce high levels of cytokines even at low concentrations, 5 μg ml[-1] purified Als3 was used in some experiments. For ELISA with BMDMs, which produce lower levels of cytokines compared to BMDCs, we usually use 15 μg ml[-1] to ensure sufficient cytokine levels for measurement.

## Immunoblotting

Hyphae were induced on plates, and then BMDMs were added into the plates at MOI 1 and cultured at 37 °C with 5% CO$_2$ for 5.5 h. After co-incubation, cells were lysed in 2x SDS loading buffer supplemented with protease inhibitor mini cOmplete (Roche), phosphatase inhibitor phosphoSTOP (Roche), and 1 mM PMSF. 293 T cells were processed as previously described[15]. In detail, 293 T cells were lysed in 20 mM Tris-HCl pH 7.5, 135 mM NaCl, 1.5 mM MgCl2, 1 mM EGTA, 1% Triton X-100, 10% glycerol, protease inhibitor (Roche) and phosphatase inhibitor (Roche). After 20 min on ice, cells were centrifuged at $20,000 \times g$ for 20 min at 4 °C, the soluble fraction was collected, and the insoluble fraction was mechanically disrupted in 6 M urea, 3% SDS, 10% glycerine, and 50 mM Tris pH 6.8. Samples were run on SDS-PAGE gels and transferred to PVDF membranes (Millipore, IPVH00010). Membranes were incubated with the following primary antibodies: anti-IL-1β (R&D Systems, AF-401-NA 1:2000), anti-caspase-1 (AdipoGen, AG-20B-0042--C100, 1:1000), anti-GSDMD (Abcam, ab209845, 1:1000), anti-β-actin-HRP (CST, 5125S, 1:5000), anti-FLAG-HRP (CST, 86861 s, 1:3000), anti-GFP-HRP (CST, 2037 s, 1:3000), anti-caspase-8 mouse-specific (Enzo, ALX-804-447-C100, 1:1000), anti-cleaved-caspase-8 mouse-specific (CST, 9429S, 1:500), anti-caspase-8 human-specific (CST, 9746S, 1:1000), anti-FADD (SCBT, sc-271748, 1:500), anti-RIPK1 (CST, 3493S, 1:1000), and anti-ASC (CST, 67824S, 1:1000). Appropriate horseradish peroxidase (HRP)-conjugated secondary antibodies (Jackson ImmunoResearch, 1:7500) were used, including anti-rabbit (111-035-047), anti-mouse (315-035-047), or anti-goat (705-035-003). Protein bands were visualized using Clarity or Clarity Max Western ECL Substrates (Bio-Rad), and membranes were developed with a Fujifilm LAS-4000 Imager; images were analyzed with ImageJ (1.51j8). Band intensity was analyzed with Image J (1.51j8). Antibodies were listed in Supplementary Table 2.

## Immunoprecipitation

His-tagged Als3 or His-tagged Als3[S170Y] were transfected into LPS-primed BMDMs (200 ng ml[-1] LPS, 2 h) with Profect-P2 transfection reagent (Targeting Systems) following the manufacturer's protocol. 1.5 h after transfection, BMDMs were lysed with a buffer containing 20 mM Tris-HCl (pH 7.4), 100 mM NaCl, 30 mM KCl, and 0.1% NP-40. The samples were centrifuged at $1900 \times g$, 4 °C for 5 min, and supernatant containing the solubilized portion was collected. The

samples were incubated with 10 μl of HisPur™ Ni-NTA magnetic beads (Thermo Scientific) overnight at 4 °C with rotation. The resin was washed with lysis buffer, and the proteins were eluted with lysis buffer containing 300 mM imidazole.

For the pull-down assay, 0.4 μg of recombinant human caspase-8 was incubated with 0.4 μg of purified Als3-His, Als3$^{S170Y}$-His, or PBS in binding buffer (HEPES, pH 7.3, 150 mM NaCl, 0.1 mM DTT) for 30 min at room temperature. The samples were then incubated with 10 μL of Pierce™ Ni-NTA magnetic agarose beads, which were pre-blocked with 5% BSA for 30 min and rotated at room temperature for an additional 30 min. Bound proteins were eluted with TBS buffer (50 mM Tris-HCl, 150 mM NaCl, pH 7.6) containing 400 mM imidazole.

293 T cells were transfected for 16 h with respective constructs and lysed as above with buffer containing 1% Triton X-100. Supernatant or whole cell lysates were used for immunoprecipitation, as described in the figures. Lysates were incubated with Anti-FLAG® M2 magnetic beads (Millipore) or Pierce™ Ni-NTA magnetic agarose beads overnight at 4 °C. The beads were washed at least 4 times with lysis buffer. Immunoprecipitation was performed according to the manufacturer's instructions.

### Cytotoxicity assay

Mouse recombinant TNF (R&D), soluble human Fc-FasL (AdipoGen), Nec-1s (CST), MCC950 (InvivoGen), and z-VAD-fmk (UBPBio) were used as indicated in the figures. Cell viability was determined using the CellTiter-Glo luminescent cell viability assay (Promega, G7571).

### Fluorescence labeling of Als3 and Als3$^{S170Y}$

Purified Als3 and Als3$^{S170Y}$were fluorescently labeled separately using Alexa Fluor® 488 Conjugation Kit (Fast) - Lightning-Link (Abcam, ab236553), according to the manufacturer's instructions. Labeled Als3, and Als3$^{S170Y}$ were separated from reagent and side products by gel-filtration chromatography using Zeba™ Dye and Biotin Removal Spin Columns (Thermo Scientific, A44296S).

### Alexa Fluor 488-conjugated Als3/Als3$^{S170Y}$ localization

Macrophages were cultured in 8-well chamber slides (Thermo Scientific) at 37 °C. For studying cell surface binding, cells were cooled to 4 °C and incubated with 15 μg ml$^{-1}$ Alexa Fluor-488 conjugated Als3/Als3$^{S170Y}$ at 4 °C for 1 h. For studying internalization, 15 μg ml$^{-1}$ Alexa Fluor-488 conjugated Als3/Als3$^{S170Y}$ was incubated with macrophages at 37 °C for 1 h. After rinsing 3 times with cold PBS, the cells were fixed with 4% paraformaldehyde (w/v in PBS) at room temperature for 5 min. The cells were permeabilized with 0.5% Triton X-100 for 5 min and blocked in TBS containing 10% FBS for 10 min at room temperature. Cells were stained with EEA1 antibody (CST, 3288 T, 1:200) or caspase-8 antibody (Enzo, ALX-804-447-C100, 1:100) at 4 °C overnight. The secondary antibody AlexaFluor647-conjugated anti-rabbit (Invitrogen) or AlexaFluor568-conjugated anti-rat (Invitrogen) was incubated with the cells for 30 min. After rinsing 3 times with TBS, an antifade mounting medium with DAPI (VECTASHIELD, H-1200-10) was used to stain the cells. Cells were visualized using a Leica SP8 confocal microscope.

### Microscopy imaging

For immunofluorescence of hypha-infected BMDMs, overnight *C. albicans* yeast culture was diluted into fresh SC medium containing 2% GlcNAc as sole carbon sources and cultured in 8-well Nunc Lab-Tek chamber slides (Thermo Scientific) at 37 °C for 2 h. After this, the medium was removed, and BMDMs were added into the chambers for interactions at MOI 1 for 2 h. Cells were then washed with chilled PBS, fixed with 4% formaldehyde for 20 min, permeabilized with 0.5% Triton X-100 for 5 min, and blocked in TBS containing 10% FBS for 10 min at room temperature. Cells were stained with unconjugated primary antibodies at 4 °C overnight as follows: anti-ASC (Millipore, 04-147,

1:100), anti-caspase-8 (Enzo, ALX-804-447-C100, 1:100), anti-Als (1: 50). Cells were then washed and incubated with secondary antibodies for 30 min as follows: FITC-conjugated anti-mouse (Jackson ImmunoResearch); AlexaFluor568-conjugated anti-rat (Invitrogen); and AlexaFluor647-conjugated anti-rabbit (Invitrogen). These antibodies were listed in Supplementary Table 2. For fluorescent imaging in 293 T cells, transfected cells were fixed with 4% paraformaldehyde (in PBS, pH 7.4) for 5 min.

To detect Als3 on the hyphal surface, hyphae induced in SC medium with 2% GlcNAc were washed with PBS and fixed with 4% formaldehyde for 20 min. The hyphae were then stained using an anti-Als antibody followed by an AlexaFluor488-conjugated anti-rabbit secondary antibody (Invitrogen).

The images were captured using a Leica SP8 confocal microscope. Image analysis was performed using Leica LAS AF software. All image data shown were representative of three independent experiments with at least six randomly selected fields for each experiment.

### Statistics and software

Statistical analysis was determined by indicated methods using GraphPad Prism software 9.5.1. The diagram was created with BioRender.

### Reporting summary

Further information on research design is available in the Nature Portfolio Reporting Summary linked to this article.

## Data availability

The source data for Figs. 1a–e, 2b, 3a, d, e, 4a, c, d, 5a–c, and 6a–c, e, f and Supplementary Figs. 1b, c, 2, 3a, b, 4, and 5a, b are provided as a Source Data file in this paper. All the plasmids, strains, and cell lines generated in this study are available from the authors upon request. All other data are available in the article and its Supplementary files or from the corresponding author upon request. Source data are provided in this paper.

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

## Acknowledgements

We thank Dr. Hamid Kashkar for *Casp8*[-/-] 293 T and its parental cell lines; Dr. Phang-Lang Chen for sharing cell-related resources; Dr. Egil Lien for providing bones from *Mlkl*[-/-], and *Mlkl*[-/-]*Casp8*[-/-] mice. This study was supported by National Institutes of Health grant R01GM117111 (H.L.), National Institutes of Health grant R01EY036478 (E.P.), and National Institutes of Health grant R01DE022600 (S.G.F.).

## Author contributions

T.T.Z. and H.L. conceived and designed the study; T.T.Z. performed most of the experiments and data analysis; N.V.S. and S.G.F. performed in vivo experiments; M.M. and E.P. assisted with getting bone marrows and other reagents; Q.Y. did part of the fungal protein purification; T.T.Z. wrote the draft; T.T.Z., H.L., S.G.F., and E.P. reviewed and edited the manuscript; H.L., E.P., and S.G.F. secured the funding.

## Competing interests

The authors declare that they have no competing interests.
