## [Transparent Peer Review file · Nature Communications]

Fungal Als proteins hijack host death effector domains to promote inflammasome signaling

Corresponding Author: Professor Haoping Liu

Version 0:

Reviewer comments:

Reviewer #1

(Remarks to the Author)

In this paper, Zhou et al investigate a moonlighting role for Als3 in directly activating immune signalling and inflammasome responses in murine macrophages. In general the work has been carried out well, but there are some conceptual issues, over-interpretation and relevance issues that need to be resolved.

Major concerns:

The authors make extensive use of a mutant for Als3 that is released (secreted) from the fungal cells and thus acts on the target host cells. How relevant is this to the real-world setting? Als3p is a cell wall protein that is not normally secreted, so on the face of it, all the work carried out with the “secreted” version is of purely academic interest and no real-world value. How do the authors match their findings to a real-world setting?

With the generated “secretion” mutant, can the authors demonstrate that there is no fungal surface bound mutant?

Were the fungal binding experiments with these mutants carried out using yeast or hyphal cells? Als3 is a hyphal expressed gene, so this is important.

What proof do the authors have that the recombinant Als3 peptides/proteins that they use are pure? What contaminants do they contain? Any PAMPs that are co-purified with the als3 protein will likely stimulate an inflammasome response.

Although the authors show some nice data demonstrating internalisation of Als3p into the mammalian cells, the activity seen could still be due to surface interactions, rather than internalisation – can the authors eliminate this possibility – i.e. using inhibitors of the surface receptors for Als3 (EGFR, Her2 etc).

Line 133 – the authors need to clarify how they know that the effects are mediated through Dectin-2 – the data says it’s only partial – what else are they predicting/demonstrating.

Line 139 – The statement that “Als3 internalisation is essential for immune responses...” is too extreme – there’s evidence for plenty of other routes for immune activation/responses that are completely independent of Als3 and its internalisation. In addition to these standard stimuli (b-glucans, mannans, candidalysin etc), blocking internalisation (i.e. with Ptistop 2) in itself might trigger a response, independent of Als3.

It's not clear to me whether the authors believe that Als3 is being internalised by Dectin-2 to activate Syk for onward activation of Caspase-8 or whether Als3 directly activates Caspase-8.

Can the authors comment on the impact of using Profect P2 to get the Als3 peptides into the cells – how can they be sure that this isn't affecting the interaction with Caspase-8? It is possible that the internal concentrations of Als3 achieved using this process are in vast excess to those naturally achievable, which in turn may skew the results due to “overloading” the system with Als3.

The co-localisation of Caspase-8 and Als3, whilst strong, does not demonstrate that the two proteins interact. The authors need to use a direct binding affinity-type assay such as the Biacore system to be able to state this. Similarly, pull-down assays are also not definitive, as other components may also be present that will not be detected in this system. Use of a system such as FRET would show direct interaction within the cell, but would be limited in demonstrating physical binding/interaction.

ALS3 gene transfection and protein production in the cell is of dubious value without extensive further structural/functional analyses. Significant difference in amino acid sequence, glycosylation etc may exist due to different codon preference, and internal cellular processes. Similarly, it may appear in cellular compartments that are physiologically irrelevant to host-fungal interactions. Several areas of work need to be done here – expression of codon-optimised Als3 in a non-infective/invasive fungal species such as *S. cerevisiae* – does this drive the same effects as seen here for *Candida*? This will answer whether other *Candida* hyphal proteins are working in concert/alongside Als3 to drive these effects in the normal physiological setting.

The authors show difference in interactions with FADD and Caspase-8. Why do they think this is occurring?

The authors should conduct similar experiments with at least some other Als proteins/mutants to demonstrate whether this phenomenon is unique to Als3 or common to Als proteins.

The concentration of IL1 β and TNF α in response to ALS3 shown in supplementary Fig1 is different from the rest of the paper - can this be explained?

General comments

Specify the MOI and time-point infection in all figure legends.

There are many abbreviations in the text that are not defined – for example Line 35 : first introduce *Candida albicans* (*C. albicans*)

Sup fig1. Are these biological replicates or just technical repeat? If its not biological replicate the experiment needs to be repeated.

Line 84: “To investigate whether the adhesion and invasion functions of Als3 determine its in vivo functionality, we deleted the GPI-anchor sequence from the ALS3 (als3GPI Δ/Δ)”. Requires more explanation, these experiments were performed using HK-2 cells - how is this an in-vivo experiment? Change statement to in-vitro.

For how long the hk-2 cells were infected and state MOI?

line 87: Mention the name of the strain in the text: SN250

Line 396: “The growth medium was changed with a fresh medium” was this serum supplemented or serum free media?

Line 542, include the post test for the statistics

Line 658 : remove the C at the end

Reviewer #2

(Remarks to the Author)

Reviewer #3

(Remarks to the Author)

The manuscript “Fungal Als proteins hijack host death effector domains to promote inflammasome signaling” presents significant findings and provides new potential targets to control hypha-induced inflammation. The manuscript is clearly presented. All tables and figures are understandable and clear. Overall, this study has a certain significance. However, I have several detailed suggestions:

Comment 1: It might be useful to add a brief description of the Als protein family in the introduction.

Comment 2: Figure 1, Figure 2 and Figure 4: The ratio of IL1RA and IL1 β is an important tool to assess the dysregulation of the immune response.

Did you measure the production of IL1Ra? And the relative IL1Ra/IL1 β ratio?

Comments 3: Several studies have documented crosstalk between cell death pathways involving NLRC4 during infections and inflammatory conditions.

For example, Man SM proposes a study in which a dynamic multiprotein complex composed of NLRC4, NLRP3, caspase-1, caspase-8 and pro-IL-1 β colocalize in the same ASC inflammasome in Salmonella-infected macrophages. The NLRC4 inflammasome can recruit caspase-8, a key component of the PANoptosome, by interacting with the PYD of ASC and the caspase-8 death effector domain.

Additionally, NLRP1b and NLRC4 trigger caspase-8-mediated apoptosis as an alternative cell death program in Casp1-deficient macrophages and intestinal epithelial organoids, providing evidence for the crosstalk between these pathways. See Sundaram B et al. *Int J Mol Sci.* 2021 Jan 21;22(3):1048.

This, along with experimental evidence showing that *P. aeruginosa*-induced cell death pathway involves key components of pyroptosis, apoptosis, and necroptosis and that MLKL-mediated cell death may be acting as a compensatory mechanism in the absence of NLRC4 (Sundaram B, Karki R, Kanneganti TD. NLRC4 Deficiency Leads to Enhanced Phosphorylation of MLKL and Necroptosis. *Immunohorizons.* 2022 Mar 17;6(3):243-252. doi: 10.4049/imunohorizons.2100118.).

Since it is well known that NLRC4 also plays a key role in *Candida* infections, have you also investigated the possible role played by NLRC4? If yes, it was suggested to be discussed.

Comment 4: The article focuses on NLRP3 inflammasome. I suggest to change the title, for instance: "Fungal Als proteins hijack host death effector domains to promote NLRP3-inflammasome signaling".

Reviewer #4

(Remarks to the Author)

Zhou et al have produced a very elegant piece of work to reveal the mechanisms underlying inflammasome activation by Als proteins of *C. albicans*. They have presented some compelling data demonstrating the first instance of a fungal virulence protein interfering with intracellular cell death signaling networks. Overall, this work is an important contribution to the field of pathogen biology and host immunity and will pave the way for further studies on fungal pathogens and how they manipulate the host to persist and disseminate. This type of work always proves valuable for the progression of our overall understanding of host immunity to infections and will be critical moving forward when considering tailored treatment plans for cancer patients or those with other chronic conditions. I have made some major and minor comments below for the authors to address. I believe this work is of high value and would be of interest to readers of *Nature Communications*, however I would like the comments sufficiently addressed before I would agree for it to be published. Really nice work by the authors, thanks for the opportunity to review the work.

Major comments

Figure 1c: Have the authors checked what else is contributing to IL-1B release? What about triple A1s mutant? Is this possible without affecting the growth of the pathogen?

Fig 1d: Is there a significant difference between als3 mutant, S170Y and V309N?

Fig 1e: There are no stats indicated between purified WT Als3 and S170Y? This would be important to mention, or included in the figure.

Fig 2a: There is no quantification of these microscopy findings? An indication of how many cells were checked would be the least amount of information expected here.

Fig 3a: Looking at this, I'm not 100% convinced there is no interaction between Als3 and NLRP3. I would say there is some amount of interaction between Als3 and NLRP3 here, certainly a little more than the control and the mutant. Can you find some explanation for why you might see some interaction there? Is there another way you could test this? Possibly via the use of NLRP3 inhibitors? Or, test IL-1B processing in the presence of Als3 in an *nlrp3* knockout cell line? I appreciate you have performed the experiment for Fig 3c, so this goes somewhat towards answering this, but how can you guarantee that casp-8/ASC association is the only mechanism at play here? After reading/looking more carefully, I see *nlrp3*^{-/-} KO cells have been used in Supp fig 3a, but this was not explained in the text. Only the inhibition of NLRP3 with KCl was explained. Mentioning what you did in supp fig 3a would support your argument that Als3 induces ASC oligomerisation via interaction with casp-8 and not NLRP3.

Supp Fig 4: why did you only use 5ug/ml of Als3 here for 24hrs, whereas in Fig 2, you used 15ug/ml for 24hrs? The IL-1B secretion in Supp fig 4 is considerably lower. Same for Fig 4, for the blots, 5ug/ml Als3 is used and for the ELISAs, 15ug/ml is used. Some explanation as to the rationale behind doses would be preferable.

Fig 4d: The way this figure is presented is very confusing. The cell viability should be represented as a percentage of 100, whereby the positive control (all cells dead), viability should be close to 0%, and healthy cells should be up around 80%. Did you correlate the Cell Titre Glo assay with any other form of cell death assay, for example LDH release assay (Cytotox by Promega)?

Fig 5: Nice experiments, however, did you perform a receptor complex (DISC) pulldown with Fc-FasL and probe for Als3? This would definitively show that Als3 was associated with the complex when induced by Fc-FasL. Also, there are no stats indicated between the vector control and N-Als3? Is this significant?

I completely understand and agree that BMDMs needed to be primed with LPS first, but did you ever try priming with live WT *C. albicans* and get a similar result with the cellular signaling you were investigating?

Do you believe that Als3 is purely binding DED or was there any evidence of PTMs mediated by the fungal protein?

Line 339: can you be a bit more explicit about N-Als3 'overlapping' with casp-8, it is not immediately clear what you mean here. If the protein sequence is identical or anything along those lines? On this, an alignment of the DED of casp-8, FADD and Als3 would be an important addition to the figures, even if supplementary. Including other DED-like proteins could also be of interest, and possibly include a couple of fungal Als proteins if appropriate? An alignment with vFLIPs and/or cFLIP may also be of use.

At the end of the discussion, the authors comment on *C. albicans* contributing to cancer prognosis by contributing to inhibition of apoptosis. I see the value in this comment, but there are many bacterial and viral pathogens that also specifically inhibit apoptosis in both epithelial and phagocytic cells, especially in the gut. This could be mentioned also to make their point, noting any references to other pathogen infections and an association with more severe outcomes in cancer patients. It would bolster their argument.

Minor comments

Line 51: The way this sentence is worded suggests the pathogen wants to induce clearance, which I am sure it does not, in fact the opposite. Could it be re-worded to say "It induces immune responses which promote fungal clearance during systemic infection". It's a small change, I think affects a different message.

Line 55: I think the cell death field would typically say 'oligomerisation' of caspase-8 rather than polymerisation.

Line 65: Fix grammar, either say 'vFLIP inhibits' or 'vFLIPs inhibit', 'vFLIPs inhibits'.

Line 87: you suddenly introduce 'the *als3* Δ/Δ mutant without clearly stating what this is. Even the following sentence is not sufficient to explain the genetic background of this mutant is. I am assuming it is a complete *als3* deletion.

Fig 1: Why have only male mice been used?

Line 120: suggests the Als3/S170Y conjugates are internalised, before the following line that also suggests they are internalised. Just check the wording and adjust please.

Line 139: You refer to 'expression' of IL-1B, and 'activation' of casp-1, it would be preferable to say IL-1B 'processing' not 'expression', this is not correct.

Line 164/165: Could be a bit more articulate about how the loss of caspase-8 mediates lethality during gestation, i.e. mention that this is largely due to 'uncontrolled' necroptosis.

Line 201: Please change RIP1 to RIPK1, it is a kinase and this is the correct annotation.

Line 320: when you say 'disclosed' what do you mean? Can this sentence be re-worded to be a bit clearer? Thank you.

Line 354-355: it is not clear what is meant by 'might impact host cell death in the long term'. Some clarity here would be good.

I am not going to knock the manuscript back because of this, but is it possible to test your in vitro work here in a mouse model and look for exacerbated IL-1B responses in serum? I am also curious whether you tested human PBMCs for IL-1B responses when exposed to *C. albicans* or Als3 in any form?

Version 1:

Reviewer comments:

Reviewer #1

(Remarks to the Author)

In their revised manuscript, Zhou et al have answered most of my comments. However, although they have shown that there is no β -Glucan contamination in their *als3* preparation, this is not the only PAMP that could contribute to a response - others include Mannans and chitin. This is not something that should prevent publication of the work, but is an issue that should be at least addressed in the text in some manner.

Reviewer #3

(Remarks to the Author)

Following the suggestions of all reviewers, changes in the initial version of the manuscript are made. I appreciate the effort made by the authors to improve the quality of the work and the time spent doing research to pose an answer to my previous questions.

In my opinion, the article can be now considered for publication.

Reviewer #4

(Remarks to the Author)

Dear Authors,

I am satisfied that you have addressed my points, thank you. I would ask that you note in the methods the reasoning you stated to me for the varied concentrations of purified Als3 used. Also, can I clarify, for my point on using live *C. albicans* to prime cells, I appreciate that this may affect downstream analyses, but is this also the case if you used 'heat-killed' *C. albicans*. As a bacteriologist we do this a lot, but I will defer to your expertise on fungal pathogens here. I just wanted to be clear about this and using the most physiologically relevant controls.

Overall, thanks for addressing my comments and good luck with the re-submission.

Jaclyn Pearson

To the reviewers:

We greatly appreciate your constructive comments and have made revisions to address them. The major changes in the revised manuscript are as follows:

1. We show in the new Supplemental Figure 1 that Als3 is present in *C. albicans* hyphal growth media, indicating that Als3 is not only on the cell wall but also could be released from the cells.
2. Fig. 5a now includes pull-down data with purified human caspase-8 and Als3 proteins to show that these proteins can directly interact.

Reviewer #1 (Remarks to the Author)

In this paper, Zhou et al investigate a moonlighting role for Als3 in directly activating immune signaling and inflammasome responses in murine macrophages. In general the work has been carried out well, but there are some conceptual issues, over-interpretation and relevance issues that need to be resolved.

Major concerns:

The authors make extensive use of a mutant for Als3 that is released (secreted) from the fungal cells and thus acts on the target host cells. How relevant is this to the real-world setting? Als3p is a cell wall protein that is not normally secreted, so on the face of it, all the work carried out with the “secreted” version is of purely academic interest and no real-world value. How do the authors match their findings to a real-world setting?

Reply: To determine if Als3 is released from wild-type *C. albicans*, we examined Als3 proteins secreted into the supernatant by WT, the *als3^{Δ/Δ}* mutant (antibody specificity control), and the *als3^{GPIΔ/Δ}* mutant (level for 100% release of Als3). The result showed that ~33% of Als3 protein from WT (SN250) was released into the supernatant (Supplementary Fig. 1a). In addition, the anti-Als antibody used was generated from the immunization with Als3 fragment (PMID: 17311474). Thus, the antibody is expected to recognize Als3 better than other Als proteins.

With the generated “secretion” mutant, can the authors demonstrate that there is no fungal surface bound mutant?

Reply: In this version, we included cell surface Als protein immunofluorescence data in Supplementary Fig. 1b. The data showed that the Als3 protein was not on the surface of the *als3^{GPIΔ/Δ}* mutant.

Were the fungal binding experiments with these mutants carried out using yeast or hyphal cells? Als3 is a hyphal expressed gene, so this is important.

Reply: The fungal binding experiments were carried out with hyphae.

What proof do the authors have that the recombinant Als3 peptides/proteins that they use are pure? What contaminants do they contain? Any PAMPs that are co-purified with the als3 protein will likely stimulate an inflammasome response.

Reply: In Supplementary Fig. 1d, we showed that only the proteins at the expected size were on the SDS-PAGE. Therefore, there is no significant contamination of protein. In addition, we performed an ELISA for detection β-glucan with the method from Invivogen (<https://www.invivogen.com/fc-mdectin-1a>). Soluble β-glucan was coated on the plate as positive controls. The minimum concentration for the positive control we used was 20 ng/ml. The β-glucan in 20 μg/ml Als3 was far below the minimum level undetectable in our assay (data not shown). Thus, we confirmed that β-glucan contamination is not an issue for our purified protein. Glycosylation on Als3 may serve as PAMP, but Als3 peptide-binding cavity is required for the inflammasome response as Als3^{S170Y} is inactive.

Although the authors show some nice data demonstrating internalisation of Als3p into the mammalian cells, the activity seen could still be due to surface interactions, rather than internalisation – can the authors eliminate this possibility – i.e. using inhibitors of the surface receptors for Als3 (EGFR, Her2 etc).

Reply: EGFR is expressed by macrophages and dendritic cells. Her2 is expressed in epithelial cells but has not been reported to be expressed in BMDMs and BMDCs. Therefore, we tested the effect of EGFR inhibitor Gefitinib on Als3-induced immune responses.

The presence of Gefitinib significantly inhibited N-Als3-, LPS-, or Pam2CSK4-mediated TNF-α production and IL-1β release. The trends are similar. Therefore, the reduction might not be due to the direct interaction of Als3-EGFR.

Figure legend: ELISA analysis of TNF- α and secreted IL-1 β from BMDCs after stimulation with purified N-Als3 (5.3 μ g ml⁻¹), LPS (5 ng ml⁻¹), or Pam2CSK4 (5 ng ml⁻¹) for 24 h with or without EGFR inhibitor Gefitinib (5 μ M). Data are mean \pm s.e.m. from two independent experiments. P values were calculated using an unpaired two-tailed t-test (n=5).

Line 133 – the authors need to clarify how they know that the effects are mediated through Dectin-2 – the data says it’s only partial – what else are they predicting/demonstrating.

Reply: Als3 is highly glycosylated as its actual size on the gel (~440-kDa) is larger than the protein molecular size ~120kDa (PMID: 10672182). Glycan staining of Als1p-N also confirmed that N-Als1p is highly glycosylated (PMID: 21585565). The contribution of Als3 protein to Dectin-2-Syk activation was shown in our recent publication (PMID: 38724513). The other parts are from CR3 (PMID: 38724513) and caspase-8 (Fig. 2-6).

Line 139 – The statement that “Als3 internalisation is essential for immune responses...” is too extreme – there’s evidence for plenty of other routes for immune activation/responses that are completely independent of Als3 and its internalisation. In addition to these standard stimuli (b-glucans, mannans, candidalysin etc), blocking internalisation (i.e. with Ptistop 2) in itself might trigger a response, independent of Als3.

Reply: We changed it to “Als3 internalization contributes to the optimal immune responses”.

It’s not clear to me whether the authors believe that Als3 is being internalised by Dectin-2 to activate Syk for onward activation of Caspase-8 or whether Als3 directly activates Caspase-8.

Reply: Als3 interacts with Dectin-2 through the α -mannans of glycosylated Als3. Whether the α -mannan/Dectin2/Syk pathway contributes to Als3 internalization is uncertain. We report in our previous publication (PMID: 38724513) that Als3 interacts with the integrin receptor CR3 on the cell surface. Integrin receptors could contribute to ligand internalization (PMID: 25600874). Therefore, we assume that several receptors may be involved in Als3 internalization.

Can the authors comment on the impact of using Profect P2 to get the Als3 peptides into the cells – how can they be sure that this isn’t affecting the interaction with Caspase-8? It is possible that the internal concentrations of Als3 achieved using this process are in vast excess to those naturally achievable, which in turn may skew the results due to “overloading” the system with Als3.

Reply: In Figure 4A, we tested the effect of Profect P2 on both Als3 and Als3S170Y, and found that Profect P2 showed an effect on Als3, but not Als3^{S170Y}, which suggests that there is no other effect of this reagent. With Profect P2, we collected the sample to analyze at 45 minutes (Fig. 4a, b). Without it, we collected the sample at 24 h (Supplementary Fig. 4a) or 48 h (Fig. 4c). therefore, Profect P2 accelerated the assay, but did not alter the results.

The co-localisation of Caspase-8 and Als3, whilst strong, does not demonstrate that the two proteins interact. The authors need to use a direct binding affinity-type assay such as the Biacore system to be able to state this. Similarly, pull-down assays are also not definitive, as other components may also be present that will not be detected in this system. Use of a system such as FRET would show direct interaction within the cell, but would be limited in demonstrating physical binding/interaction.

Reply: Human caspase-8 can self-aggregate from the DED-DED interaction. Protein aggregates will interfere with SPR and can damage the sensor or integrated flow chamber. We have now included pull-down data with purified proteins to show the interaction between human caspase-8 and Als3 (Fig. 5a).

ALS3 gene transfection and protein production in the cell is of dubious value without extensive further structural/functional analyses. Significant difference in amino acid sequence, glycosylation etc may exist due to different codon preference, and internal cellular processes. Similarly, it may appear in cellular compartments that are physiologically irrelevant to host-fungal interactions. Several areas of work need to be done here – expression of codon-optimised Als3 in a non-infective/invasive fungal species such as *S. cerevisiae* – does this drive the same effects as seen here for *Candida*? This will answer whether other *Candida* hyphal proteins are working in concert/alongside Als3 to drive these effects in the normal physiological setting.

Reply: We demonstrated that Als3's role in inducing inflammasome activation is more pronounced in fast-growing, GlcNAc-induced hyphae compared to the relatively slower-growing RPMI-induced hyphae (PMID: 38724513, Fig. 3 b, c). Therefore, we hypothesize that Als3 on *S. cerevisiae* may not be as effective. We confirmed that GlcNAc-induced hyphae release a substantial amount of Als3; however, whether Als3 can also be released from the yeast form remains unknown.

The authors show difference in interactions with FADD and Caspase-8. Why do they think this is occurring?

Reply: We state in the text on page 8 that 'Cryo-EM structural analysis showed that caspase-8 filament formation is nucleated by FADD via DED interactions (PMID: 33547302), as evidenced by the sequential assembly of FADD short filaments followed by caspase-8 long filaments. Since N-Als3 completely overlaps with caspase-8, we speculate that it might share more features with the DEDs of caspase-8 compared to the DED of FADD'.

The authors should conduct similar experiments with at least some other Als proteins/mutants to demonstrate whether this phenomenon is unique to Als3 or common to Als proteins.

Reply: *Ex vivo* experiments showed that the *als1^{Δ/Δ}* mutant and the *als5^{Δ/Δ}* mutant also had defects in inducing IL-1 β (Fig. 1c). Therefore, we assume that this might be a common feature of Als proteins. We state in the Discussion that future studies will examine other members of the Als family proteins.

The concentration of IL1 β and TNF α in response to ALS3 shown in Supplementary Fig1 is different from the rest of the paper - can this be explained?

Reply: In Supplementary Fig1, we used BMDCs to analyze cytokines in 24-well plates with smaller medium volumes. This was our initial experiment examining Als-induced cytokines. Later on, to save protein, we changed the protocols to 96-well plates. After receiving this comment, we repeated the experiment in 96-well plates with BMDCs from two mice. We found that the maximum IL-1 β level could be 526 pg/ml (not shown), which is lower than the data we used (~820 pg/ml). To be consistent, we deleted the old data that was done in 24-well plates.

In addition, BMDCs generate more cytokines than BMDMs, which we used in some experiments in this study.

General comments

Specify the MOI and time-point infection in all figure legends.

Reply: We only used one condition for all infections, which was described in the "Methods" section "Infection or stimulation of BMDMs and BMDCs". The condition is MOI 1, 24h for BMDCs and 5.5h for BMDMs.

There are many abbreviations in the text that are not defined – for example Line 35 : first introduce *Candida albicans* (*C. albicans*)

Reply: Changed. We also added the full names of FADD and cFLIP in the introduction. Thanks.

Sup fig1. Are these biological replicates or just technical repeat? If its not biological replicate the experiment needs to be repeated.

Reply: We included the results from another experiment. Thus, two independent experiments from biological replicates are shown.

Line 84: "To investigate whether the adhesion and invasion functions of Als3 determine its in vivo functionality, we deleted the GPI-anchor sequence from the ALS3 (*als3GPI Δ / Δ*)". Requires more explanation, these experiments were performed using HK-2 cells - how is this an in-vivo experiment? Change statement to in-vitro.

Reply: Sorry for the confusion. We reorganized the sentences. “We deleted the GPI-anchor sequence from the *ALS3* (*als3^{GPIA/A}*), resulting in a mutant that secretes Als3 (Supplementary Fig. 1a, b) and has reduced adhesion (Supplementary Fig. 1c). To investigate whether the adhesion and invasion functions of Als3 determine its *in vivo* functionality, we infected mice with the wild-type SN250 (WT), the *als3^{A/A}* mutant (full-length *als3* coding sequence knockout in WT⁹), and the *als3^{GPIA/A}* mutant.”

For how long the hk-2 cells were infected and state MOI?

Reply: Thanks for asking. The infection number and time are shown in the method part. It is not counted by MOI. 330 *Candida* cells were used for each well. The co-culture time was 3 hours.

line 87: Mention the name of the strain in the text: SN250

Reply: Added, thanks.

Line 396: “The growth medium was changed with a fresh medium” was this serum supplemented or serum free media?

Reply: The medium is still DMEM/F12 (Gibco) supplemented with 10% FBS (Corning), 1% penicillin-streptomycin (Gibco). We added “complete” in the sentence to help readers understand this. “The growth medium was changed with a fresh complete medium 1 day prior to the adhesion assay.”

Line 542, include the post test for the statistics

Reply: In the figure legend, we showed detailed statistics and post-analysis methods.

Line 658 : remove the C at the end

Reply: Removed, thanks.

Reviewer #2 (Remarks to the Author)

Reviewer #3 (Remarks to the Author)

The manuscript “Fungal Als proteins hijack host death effector domains to promote inflammasome signaling” presents significant findings and provides new potential targets to control hypha-induced inflammation. The manuscript is clearly presented. All tables and figures are understandable and clear. Overall, this study has a certain significance. However, I have several detailed suggestions:

Comment 1: It might be useful to add a brief description of the Als protein family in the introduction.

Reply: This is a brief description in the second paragraph.

Comment 2: Figure 1, Figure 2 and Figure 4: The ratio of IL1Ra and IL1 β is an important tool to assess the dysregulation of the immune response.

Did you measure the production of IL1Ra? And the relative IL1Ra/IL1 β ratio?

Reply: After receiving these comments, we did a literature study about IL1Ra and its role in *C. albicans*-mediated inflammation. So far, no publication has studied the effect of IL-1Ra on the immune regulation of *C. albicans*-induced IL-1 β . Instead, other authors are enthusiastic about IL-1 β release. The processing of pro-IL-1 β to mature IL-1 β (p17) and the release of mature IL-1 β is dependent on inflammasome activation (cell death/pore formation) in BMDMs and BMDCs. Therefore, IL-1 β processing serves as a marker of inflammasome activation.

We measured IL-1Ra from BMDCs after Als3, Als3S170Y, or medium treatment. We found that Als3, but not Als3S170Y, induced a high level of IL-1Ra (data not shown). We think the data about IL-1Ra may distract readers from inflammasome activation to IL-1 β 's activity.

Comments 3: Several studies have documented crosstalk between cell death pathways involving NLRC4 during infections and inflammatory conditions.

For example, Man SM proposes a study in which a dynamic multiprotein complex composed of NLRC4, NLRP3, caspase-1, caspase-8 and pro-IL-1 β colocalize in the same ASC inflammasome in Salmonella-infected macrophages. The NLRC4 inflammasome can recruit caspase-8, a key component of the PANoptosome, by interacting with the PYD of ASC and the caspase-8 death effector domain.

Additionally, NLRP1b and NLRC4 trigger caspase-8-mediated apoptosis as an alternative cell death program in Casp1-deficient macrophages and intestinal epithelial organoids, providing evidence for the crosstalk between these pathways. See Sundaram B et al. Int J Mol Sci. 2021 Jan 21;22(3):1048.

This, along with experimental evidence showing that *P. aeruginosa*-induced cell death pathway involves key components of pyroptosis, apoptosis, and necroptosis and that MLKL-mediated cell death may be acting as a compensatory mechanism in the absence of NLRC4 (Sundaram B, Karki R, Kanneganti TD. NLRC4 Deficiency Leads to Enhanced Phosphorylation of MLKL and Necroptosis. Immunohorizons. 2022 Mar 17;6(3):243-252. doi: 10.4049/immunohorizons.2100118.).

Since it is well known that NLRC4 also plays a key role in *Candida* infections, have you also investigated the possible role played by NLRC4? If yes, It was suggested to be discussed.

Reply: Thanks for bringing up NLRC4 and the relevant references.

NLRC4 plays an important role in limiting mucosal candidiasis when functioning at the level of the mucosal stroma (PMID: 22174673). NLRC4 is also activated in vaginal candidiasis, where epithelial NLRC4 is responsible for sensing *Candida* (PMID: 26269955). In BMDMs, NLRC4 doesn't play a role in pyroptosis after *C. albicans* infection in BMDMs (PMID: 24376002). Since we mainly used BMDMs and BMDCs for experiments, the role of NLRC4 is weak. Als3 may also affect the activation of NLRC4 in mucosal or vaginal cells, which might be interesting to check in the future. However, in this manuscript, we think that we don't have adequate information to link our study with NLRC4.

Comment 4: The article focuses on NLRP3 inflammasome. I suggest to change the title, for instance: "Fungal Als proteins hijack host death effector domains to promote NLRP3-inflammasome signaling".

Reply: Since Als3 affects ASC polymerization through DEDs. ASC functions as the central adaptor for inflammasome assembly. Therefore, we think the inflammasome is more conservative than the NLRP3 inflammasome in the title.

Reviewer #4 (Remarks to the Author):

Zhou et al have produced a very elegant piece of work to reveal the mechanisms underlying inflammasome activation by Als proteins of *C. albicans*. They have presented some compelling data demonstrating the first instance of a fungal virulence protein interfering with intracellular cell death signaling networks. Overall, this work is an important contribution to the field of pathogen biology and host immunity and will pave the way for further studies on fungal pathogens and how they manipulate the host to persist and disseminate. This type of work always proves valuable for the progression of our overall understanding of host immunity to infections and will be critical moving forward when considering tailored treatment plans for cancer patients or those with other chronic conditions. I have made some major and minor comments below for the authors to address. I believe this work is of high value and would be of interest to readers of Nature Communications, however I would like the comments sufficiently addressed before I would agree for it to be published. Really nice work by the authors, thanks for the opportunity to review the work.

Major comments

Figure 1c: Have the authors checked what else is contributing to IL-1 β release? What about triple Als mutant? Is this possible without affecting the growth of the pathogen?

Reply: In our previous publication (PMID: 38724513), we screened *C. albicans* mutants with individual deleted genes encoding hyphal surface or secreted proteins and found that the *als3^{Δ/Δ}* mutant is the only one consistently showing a low ability to induce IL-1 β .

In the previous publication (PMID: 38724513), we showed that "The *als1^{S170Y}als3^{S170Y}als5^{Δ/Δ}* (*als1^{SY}als3^{SY}als5^{Δ/Δ}* in short) triple mutant induced lower level of IL-1 β compared to the *als3^{S170Y}als5^{Δ/Δ}* double mutant (Supplementary Fig. 1d)."

We observed the hyphal length each time after hyphal induction and didn't find the *als* mutants had growth defect. One example that shows WT and *als1^{S170Y}als3^{S170Y}als5^{Δ/Δ}* have similar lengths is in Supplementary Fig. 4 (PMID: 38724513).

Fig 1d: Is there a significant difference between *als3* mutant, S170Y and V309N?

Reply: We added statistical analysis to them. The *als3^{Δ/Δ}* mutant is significantly different from the *als3^{v309N}* mutant ($P=0.0369$).

Fig 1e: There are no stats indicated between purified WT Als3 and S170Y? This would be important to mention, or included in the figure.

Reply: Added, thanks.

Fig 2a: There is no quantification of these microscopy findings? An indication of how many cells were checked would be the least amount of information expected here.

Reply: We added a sentence in the figure legend to describe this. "At least 100 cells were checked in each experiment and the pictures are representative."

Fig 3a: Looking at this, I'm not 100% convinced there is no interaction between Als3 and NLRP3. I would say there is some amount of interaction between Als3 and NLRP3 here, certainly a little more than the control and the mutant. Can you find some explanation for why you might see some interaction there? Is there another way you could test this? Possibly via the use of NLRP3 inhibitors? Or, test IL-1B processing in the presence of Als3 in an *nlrp3* knockout cell line? I appreciate you have performed the experiment for Fig 3c, so this goes somewhat towards answering this, but how can you guarantee that casp-8/ASC association is the only mechanism at play here? After reading/looking more carefully, I see *nlrp3*^{-/-} KO cells have been used in Supp fig 3a, but this was not explained in the text. Only the inhibition of NLRP3 with KCl was explained. Mentioning what you did in supp fig 3a would support your argument that Als3 induces ASC oligomerisation via interaction with casp-8 and not NLRP3.

Reply: In this revised manuscript, we used "weak interactions" to describe the interaction of Als3 and NLRP3. Als3/caspase-8 interaction promotes ASC oligomerization (Fig. 3e). Since ASC is an important component of the NLRP3 inflammasome and we showed that Als3-mediated NLRP3 inflammasome activation in Fig 4, it was not a surprise to see NLRP3 in Als3 pull-down experiment.

We think Fig.3e provides strong data that shows that NLRP3 is not a direct interacting protein of Als3 in this complex. HEK293T doesn't express NLRP3. In HEK293T cells, we still can see Als promotes ASC oligomerization, which is caspase-8 dependent (Fig. 3e).

Supp Fig 4: why did you only use 5ug/ml of Als3 here for 24hrs, whereas in Fig 2, you used 15ug/ml for 24hrs? The IL-1B secretion in Supp fig 4 is considerably lower. Same for Fig 4, for the blots, 5ug/ml Als3 is used and for the ELISAs, 15ug/ml is used. Some explanation as to the rationale behind doses would be preferable.

Reply: Thanks for pointing this out. From our experience, the protein transfection experiment doesn't need a high concentration of protein. For ELISA with BMDCs, the cells make high levels of cytokines even at low concentrations. Therefore, to save proteins, we use 5 ng/ml in some experiments. For ELISA with BMDMs, the cells make lower levels of cytokines compared to BMDCs. Thus, we tend to use 15 ug/ml to make sure that we get cytokines each time.

Fig 4d: The way this figure is presented is very confusing. The cell viability should be represented as a percentage of 100, whereby the positive control (all cells dead), viability should be close to 0%, and healthy cells should be up around 80%. Did you correlate the Cell Titre Glo assay with any other form of cell death assay, for example LDH release assay (Cytotox by Promega)?

Reply: Sorry for the confusion. Since this assay is from 48 hours' cultures. LDH is not stable over a long time. So, it is not possible to get an accurate result with the LDH assay. To avoid the confusion, we changed the label as "relative cell viability".

Fig 5: Nice experiments, however, did you perform a receptor complex (DISC) pulldown with Fc-FasL and probe for Als3? This would definitively show that Als3 was associated with the complex when induced by Fc-FasL. Also, there are no stats indicated between the vector control and N-Als3? Is this significant? I completely understand and agree that BMDMs needed to be primed with LPS first, but did you ever try priming with live WT *C. albicans* and get a similar result with the cellular signaling you were investigating? Do you believe that Als3 is purely binding DED or was there any evidence of PTMs mediated by the fungal protein?

Reply: We didn't treat the HEK-cells with Fc-FasL in Fig. 5b. The data from Fig. 3 indicates that the interaction between Als3 and caspase-8 shouldn't be dependent on death ligand, for example, FasL. To induce apoptosis, we did treat Jurkat cells with Fc-FasL (Fig. 5b, now Fig. 5c).

The statistic indication of vector and N-Als3 is shown in the Figure and it is significant.

We did infect BMDMs with WT and mutant *C. albicans*, for example, Fig. 3d. But we didn't prime cells with *C. albicans*. If priming with LPS, we can easily remove the supernatant and perform a later experiment. However, if the macrophages were primed with *C. albicans*, the pathogens would be engulfed and affect later experiments.

We believe that there is a protein-protein interaction. In Fig. 6, we expressed N-Als3 in HEK293T cells by transfecting a plasmid expressing N-Als3, we still can see that N-Als3 promotes caspase-8 oligomerization. Thus, this is a process that doesn't depend on fungal PTMs.

Line 339: can you be a bit more explicit about N-Als3 'overlapping' with casp-8, it is not immediately clear what you mean here. If the protein sequence is identical or anything along those lines? On this, an alignment of the DED of casp-8, FADD and Als3 would be an important addition to the figures, even if supplementary. Including other DED-like proteins could also be of interest, and possibly include a couple of fungal Als proteins if appropriate? An alignment with vFLIPs and/or cFLIP may also be of use.

At the end of the discussion, the authors comment on *C. albicans* contributing to cancer prognosis by contributing to inhibition of apoptosis. I see the value in this comment, but there are many bacterial and viral pathogens that also specifically inhibit apoptosis in both epithelial and phagocytic cells, especially in the gut. This could be mentioned also to make their point, noting any references to other pathogen infections and an association with more severe outcomes in cancer patients. It would bolster their argument.

Reply: We changed the "overlapping" to "colocalizes" to make it more clear.

We used BLAST @-blastp to analyze the sequence similarity between Als3 and mouse caspase-8 or FADD. The results showed "no significant similarity found" for the similarity with caspase-8, while only 3% of the FADD sequence could align with Als3 with a 33.33% percentage identity (see below). Therefore, we think their protein sequences are distinct.

Description	Scientific Name	Max Score	Total Score	Query Cover	E value	Per. Ident	Acc. Len	Accession
[x] FADD		21.9	43.9	3%	0.044	33.33%	205	Query_1924471

We added one sentence to point out that many bacteria and viruses also inhibit apoptosis. "Some viral and bacterial infections also inhibit host cell apoptosis." We have mentioned that Kaposi sarcoma herpesvirus vFLIP promotes cancer.

Minor comments

Line 51: The way this sentence is worded suggests the pathogen wants to induce clearance, which I am sure it does not, in fact the opposite. Could it be re-worded to say "It induces immune responses which promote fungal clearance during systemic infection". It's a small change, I think affects a different message.

Reply: Changed, thanks for pointing this out.

Line 55: I think the cell death field would typically say 'oligomerisation' of caspase-8 rather than polymerisation.

Reply: Changed all of them, thanks.

Line 65: Fix grammar, either say 'vFLIP inhibits' or 'vFLIPs inhibit', 'vFLIPs inhibits'.

Reply: Changed to 'vFLIPs inhibit', thanks.

Line 87: you suddenly introduce 'the *als3*Δ/Δ mutant without clearly stating what this is. Even the following sentence is not sufficient to explain the genetic background of this mutant is. I am assuming it is a complete *als3* deletion.

Reply: We added a description after the mutant, "the *als3*^{Δ/Δ} mutant (full-length *als3* coding sequence knockout in WT⁹)".

Fig 1: Why have only male mice been used?

Reply: In the "Methods", we added several sentences to explain this. Both male and female mice were used in the systemic infection experiments. However, there was no significant difference in survival rates between females infected with the WT strain and those infected with the *als3*^{Δ/Δ} mutant. As a result, the survival data for females was not included in the analysis.

Line 120: suggests the Als3/S170Y conjugates are internalised, before the following line that also suggests they are internalised. Just check the wording and adjust please.

Reply: Changed into "After incubating at 37°C for 15 or 30 minutes, conjugated Als3 and Als3^{S170Y} were located in the cytosolic organelles (Fig. 2a).".

Line 139: You refer to ‘expression’ of IL-1B, and ‘activation’ of casp-1, it would be preferable to say IL-1B ‘processing’ not ‘expression’, this is not correct.

Reply: Changed, thanks.

Line 164/165: Could be a bit more articulate about how the loss of caspase-8 mediates lethality during gestation, i.e. mention that this is largely due to ‘uncontrolled’ necroptosis.

Reply: Changed into “Mice lacking caspase-8 experience embryonic lethality during gestation due to uncontrolled necroptosis, a form of regulated cell death³⁸. However, they survive beyond weaning when either of the necroptosis-mediating genes, *Rip3* or *Mkl1*, is co-ablated, preventing the excessive necroptotic response³⁸. ”

Line 201: Please change RIP1 to RIPK1, it is a kinase and this is the correct annotation.

Reply: Changed, thanks.

Line 320: when you say ‘disclosed’ what do you mean? Can this sentence be re-worded to be a bit clearer? Thank you.

Reply: Changed “disclosed” into “discovered”.

Line 354-355: it is not clear what is meant by ‘might impact host cell death in the long term’. Some clarity here would be good.

Reply: We deleted “in the long term”.

I am not going to knock the manuscript back because of this, but is it possible to test your in vitro work here in a mouse model and look for exacerbated IL-1B responses in serum? I am also curious whether you tested human PBMCs for IL-1B responses when exposed to *C. albicans* or Als3 in any form?

Reply: In our previous publication (PMID: 38724513), we measured IL-1 β in kidneys in a systemic infection model. “In comparison with mice infected with WT *C. albicans*, those infected with the *als3^{Δ/Δ}* mutant showed a trend towards reduced cytokines and chemokines, including IL-1 β , TNF- α , IL-6, IL-10, CXCL1/KC, and CXCL2/MIP-2 α , with only the reduction of TNF- α being statistically significant (Fig. 2b and Supplementary Fig. 2b). In contrast, the *als1^{SY}als3^{SY}als5^{Δ/Δ}* strain induced significantly lower levels of all these cytokines and chemokines. Thus, the Als family is required for optimal immune responses.”

We greatly appreciate the reviewers' positive feedback on our revisions and their additional suggestions for improvement.

Reviewer #1 (Remarks to the Author):

In their revised manuscript, Zhou et al have answered most of my comments. However, although they have shown that there is no b-Glucan contamination in their als3 preparation, this is not the only PAMP that could contribute to a response - others include Mannans and chitin. This is not something that should prevent publication of the work, but is an issue that should be at least addressed in the text in some manner.

Reply: We added a paragraph in discussion to further address the key role of PBC in Als3-induced immune responses, while we brought up the data with Dectin-2^{-/-} to address the contribution of glycosylation.

“The deficiency of Dectin-2 led to a decrease in N-Als3-induced IL-1 β release in BMDCs (Supplementary Fig. 2), indicating that the mannans of N-Als3 contribute to its activity. However, the mutation of PBC completely abolished the activity of Als3 in inducing immune responses, although the mutant protein Als3^{S170Y} still retains glycosylation. This suggests that protein structure determines whether Als3 is active, while glycosylation contributes to the level of activity.”

Reviewer #3 (Remarks to the Author):

Following the suggestions of all reviewers, changes in the initial version of the manuscript are made. I appreciate the effort made by the authors to improve the quality of the work and the time spent doing research to pose an answer to my previous questions.

In my opinion, the article can be now considered for publication.

Thanks for your support.

Reviewer #4 (Remarks to the Author):

Dear Authors,

I am satisfied that you have addressed my points, thank you. I would ask that you note in the methods the reasoning you stated to me for the varied concentrations of purified Als3 used. Also, can I clarify, for my point on using live *C. albicans* to prime cells, I appreciate that this may affect downstream analyses, but is this also the case if you used 'heat-killed' *C. albicans*. As a bacteriologist we do this a lot, but I will defer to your expertise on fungal pathogens here. I just wanted to be clear about this and using the most physiologically relevant controls.

Overall, thanks for addressing my comments and good luck with the re-submission.

Jaclyn Pearson

Dear Dr. Pearson,

Thanks for your comments.

We added the explanation to the method of Cytokine ELISA.

“Cytokine levels were determined using mouse IL-1 β , TNF- α , and IL-23 uncoated ELISA kits (Invitrogen) according to the manufacturer's instructions. For ELISA with BMDCs, which produce high levels of cytokines even at low concentrations, 5 $\mu\text{g ml}^{-1}$ purified Als3 was used in some experiments. For ELISA with BMDMs, which produce lower levels of cytokines compared to BMDCs, we usually use 15 $\mu\text{g ml}^{-1}$ to ensure sufficient cytokine levels for measurement.”

Live germinating hyphae induce much higher levels of IL-1 β release than heat-killed *C. albicans* yeast or hyphae. We addressed this in our recent publication PMID:38724513.